# The relationship between musical training and the processing of audiovisual correspondences: Evidence from a reaction time task

Riku Ihalainen[1,2]*, Georgios Kotsaridis[3], Ana B. Vivas[3], Evangelos Paraskevopoulos[2,4]

1 School of Computing, University of Kent, Canterbury, United Kingdom, 2 Psychology Department, City College, University of Sheffield, International Faculty, Thessaloniki, Greece, 3 Department of Psychology, CITY College, University of York Europe Campus, Thessaloniki, Greece, 4 Department of Psychology, University of Cyprus, Nicosia, Cyprus

* riku.ihalainen@outlook.com

**Data Availability Statement:** The data are publically accessible and can be retrieved from https://gin.g-node.org/rihalai/Crossmodal_Correspondences.

## Abstract

Numerous studies have reported both cortical and functional changes for visual, tactile, and auditory brain areas in musicians, which have been attributed to long-term training induced neuroplasticity. Previous investigations have reported advantages for musicians in multisensory processing at the behavioural level, however, multisensory integration with tasks requiring higher level cognitive processing has not yet been extensively studied. Here, we investigated the association between musical expertise and the processing of audiovisual crossmodal correspondences in a decision reaction-time task. The visual display varied in three dimensions (elevation, symbolic and non-symbolic magnitude), while the auditory stimulus varied in pitch. Congruency was based on a set of newly learned abstract rules: "The higher the spatial elevation, the higher the tone", "the more dots presented, the higher the tone", and "the higher the number presented, the higher the tone", and accuracy and reaction times were recorded. Musicians were significantly more accurate in their responses than non-musicians, suggesting an association between long-term musical training and audiovisual integration. Contrary to what was hypothesized, no differences in reaction times were found. The musicians' advantage on accuracy was also observed for rule-based congruency in seemingly unrelated stimuli (pitch-magnitude). These results suggest an interaction between implicit and explicit processing–as reflected on reaction times and accuracy, respectively. This advantage was generalised on congruency in otherwise unrelated stimuli (pitch-magnitude pairs), suggesting an advantage on processes requiring higher order cognitive functions. The results support the notion that accuracy and latency measures may reflect different processes.

## Introduction

Human beings are equipped with multiple sensory channels allowing us to produce a unified and coherent representation of the outside world. Recent evidence suggests that multimodal

**Funding:** This project has received funding from the Hellenic Foundation for Research and Innovation (HFRI) and the General Secretariat for Research and Technology (GSRT), under grant agreement No [2089]. The funders had no role in study design, data collection and analysis, decision to publish, or preparation of the manuscript.

**Competing interests:** The authors have declared that no competing interests exist.

sensory processing occurs at an early processing stage, and is cortically widespread (for a review, see [1]). Studies also suggest that specific life experiences may lead to enhanced multi-modal sensory processing. For instance, professional musicians appear to have extensive experience in multisensory processing, since reading, interpreting, and acting on musical notation combines at least visual, auditory, and motor information. Similarly, playing an instrument requires simultaneous processing of stimuli from at least visual, auditory, and tactile sensory modalities [2, 3]. At the cortical level, research has shown that musicians and non-musicians have both cortical and functional differences, often hypothesised to long-term training induced neuroplasticity [e.g. 4–10].

Crossmodal correspondence refers to the mapping that the observer expects between two or more seemingly arbitrary stimuli from different modalities inducing congruency effects in performance [11–13]. For example, people tend to naturally associate higher pitch with smaller objects, and with objects that are higher in spatial elevation [14, 15]. Similarly, lower pitch is typically associated with larger objects and with objects lower in spatial elevation [16]. Recent evidence suggests that our brain automatically integrates stimuli based on parameters such as temporal or spatial proximity [2, 15, 17], previous experience [18], innate cross modal correspondences and statistics of natural scenes [19, 20].

Furthermore, a number of studies have successfully used newly learned abstract rules in order to induce audiovisual congruency effects in musicians and non-musicians. Paraskevopoulos et al. [21] investigated the cortical responses (EEG/MEG) of musicians and non-musicians in relation to congruent and incongruent audiovisual magnitude comparisons of symbolic nature. The judgements were made based on a newly learned abstract rule: 'the higher the pitch, the larger the number presented'. Their results indicated two distinct neural networks for congruent and incongruent comparisons: frontotemporal and occipital areas in congruent condition, and temporal and parietal regions in the incongruent condition. Musicians further performed better—had higher accuracy—at discriminating whether the audiovisual stimuli were congruent with the rule, suggesting that musical expertise may be associated to enhanced processing of audiovisual stimuli. Similar results were found with audiovisual stimuli varying in spatial elevation in conjunction with pitch [22], see also [23], and later, with stimuli with the visual dimension representing different magnitudes in conjunction with varying pitch [24].

The evidence thus suggests that long-term training-induced neuroplasticity in musicians' is associated with advantages that can be explicitly measured with both, neuroimaging methods and behavioural measures. However, these studies have employed only precision (e.g. accuracy of discrimination judgments) as the behavioural measure. As two commonly applied behavioural measures, accuracy and reaction time have been argued to tap onto different underlying processes, with accuracy reflecting more explicit and reaction time more implicit processing [25–28], see also [29]. Hence, the two measures provide complementary information, which we aim to capture by incorporating both of the measures.

There is a relatively large number of reaction time studies done with non-musicians in the context of multimodal sensory integration. These investigations have consistently reported better performance with multimodal stimuli: faster detection of the target in the multimodal condition relative to unimodal condition (see for example [14]; for reviews see [12, 13]). Moreover, some previous studies have suggested a processing speed advantage with multimodal stimuli for musicians over non-musicians. For instance, Laundry and Champoux [30] examined auditory, tactile, and audio-tactile processing with a detection reaction time task, in which the participants were instructed to click a mouse button immediately upon perception of auditory, tactile, or simultaneous audio-tactile stimuli, and reported that long-term musical training resulted in faster response times for all the three stimulus types. Bidelman [31] investigated the

effects of music training on the temporal binding of audiovisual multisensory stimuli using the double-flash illusion [32, 33]. In the illusion, a single flash of a visual stimulus is perceived as two separate visual stimuli; an effect caused by two consecutive auditory signals (beeps) presented concurrently with the visual stimulus. Their results indicated that musicians, in comparison to non-musicians, were more accurate and faster in indicating whether a single or double flash was presented. Further, the temporal window in which the illusory effect was successfully induced was 2–3 times shorter for musicians than it was for non-musicians. Taken together, experience-induced plasticity effects on musicians seemed to extend beyond simple listening skills, leading to a more refined, improved integration of multiple sensory systems in a domain-general manner.

Earlier studies have also investigated the associations between magnitudes, pitch, and spatial elevation in the context of spatial mapping. Spatial Numerical Association of Response Codes (SNARC; [34]) and Spatial Musical Association of Response Codes (SMARC–also known as SPARC, Spatial Pitch Association of Response Codes; [35]) refer to faster response times when using the left hand responses to discriminate space-related interactions between smaller numbers and pitch. Right hand responses, on the other hand, show a similar advantage for larger numbers. This phenomenon seems to affect both speed and accuracy of responses, having a preferential pairing with the response location (upper and/or right for higher pitch), respectively. Such associations between perception and action/response are believed to be the result of an automatism [36].

Such audiovisual correspondences may also occur at an early processing level; they are thought to represent pre-attentive processing that is largely correlated with sub-cortical structures such as the superior colliculi ([37]; for a review see [38]). In contrast, magnitude judgements are thought to require higher cognitive processes, and seem to be associated with cortical sources such as the posterior parietal cortex, or the intra-parietal sulcus [21, 39]. By exploring the effects of multimodal musical training on multimodal integration in both magnitude judgements and pitch-elevation correspondence, we aim to investigate the potential link between implicit and explicit cognitive processes. In doing so, we analyse both, the response latencies (thought to reflect more implicit processing) and accuracy (thought to reflect more explicit processing; [26]). In an attempt to correct for the possible speed-accuracy trade-off effects, we furthermore analyse the combined linear integrated speed-accuracy scores (LISAS; [40–42]).

Hence, we employ a decision reaction time task, based on abstract rules related to audiovisual stimuli in in a group of musicians and non-musicians. Grounded on previous research [12–14, 21, 22, 30, 31], we hypothesize that the overall performance will be better (faster and more accurate responses) in the congruent condition relative to the incongruent condition for both groups reflecting integration of the audio- and visual stimulus. Crucially, we expect that musicians will have an overall better performance (faster and more accurate) than non-musicians due to training-induced enhancements in multisensory integration.

## Materials and methods

### Participants

Following the power analysis (reported below in the Analysis section), data were collected from 27 musicians and 23 non-musicians. One musician who was left-handed and one control participant, who had weekly piano lessons between the ages of 7 to 12 years old, were excluded from the study. Thus, the dataset consisted of 48 right-handed participants (26 musicians and 22 non-musicians). Data were further trimmed based on reaction time outliers (see Analysis section below) after which we were left with a final sample of 44 participants (25 musicians; 24 females).

Musicians (mean age = 31.40 years, SD = 12.26, range: 20–64 years, 7 females) were recruited through social media sites and via email. A musician was defined with the minimum criteria of having at least 3 years of musical education (including possible careers/music teaching) in addition to compulsory music lessons in elementary and junior high school, being currently active, and having an ability to read musical notes. The mean formal musical education (in years) amongst the musicians was 15.4 (SD = 9.4) with a range of 3–41 years. Only 2 musicians reported having less than 6 years of training. Instruments played by the musicians were considerably heterogeneous; most reported instruments were guitar (N = 6) and piano (N = 4).

Non-musicians (mean age = 28.47 years, SD = 7.19, range: 19–47 years, 11 females; mean ages did not significantly differ between musicians and non-musicians, $p > .05$) were recruited mainly via social media sites and email, and were defined as having no formal musical training in addition to the music lessons compulsory in elementary and junior high school. All of the non-musician participants also self-identified as having no musical expertise. Any continuous lessons or practicing with an instrument resulted in an exclusion from the study.

None of the participants had a history of brain trauma or mental health issues, they all had normal or corrected-to-normal vision and normal auditory thresholds/hearing, and identified as right-handed. The study was approved by the University of Sheffield Ethics committee, and was conducted in accordance with the Declaration of Helsinki (1973).

## Stimuli

Each pair of audiovisual stimuli consisted of a visual and an auditory part. The visual part of the stimulus may be encountered by one of three different types of attributes: spatial elevation, symbolic magnitude, and non-symbolic magnitude. Spatial elevation consisted of five white horizontal lines against a black background, similar to the staff lines in musical notation. A blue dot (about 2 cm in diameter, RGB colour codes: red, 86; green, 126; and blue, 214) was then placed into one of the 4 spaces between the lines, thus forming 4 different images in which the blue dot varied in spatial elevation. The colour blue was chosen for all stimuli as it has been suggested not to have any natural association with common sensory attributes [11].

The symbolic magnitude condition consisted of a number written in Calibri font in blue colour against a black background, with numbers varying from 1 to 4. The non-symbolic magnitude condition consisted of blue dots (varying in number from one to four, and of the same size and colour as in spatial elevation category) projected against a black background (see Fig 1). Each visual stimulus type was then combined with a sinusoidal tone (44,100 kHz, 16 bit) lasting for 400 ms including a 10-ms rise and decay time, thus forming stimuli displays that consisted of both an auditory and a visual modality. The tones were adopted from Paraskevopoulos et al. [21]; F5, 698.46 Hz; A5, 880.46 Hz; C6, 1046.50 Hz; or E6, 1318.51 Hz. The sounds

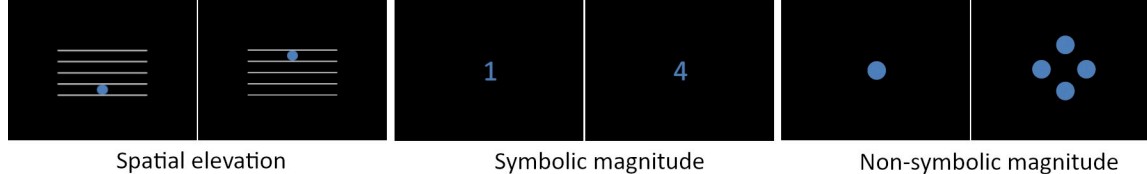

Spatial elevation Symbolic magnitude Non-symbolic magnitude

**Fig 1. Examples of each of the three stimulus categories.** Such pairs were used to form the experimental stimuli; both images in all three of the stimulus categories were paired (within-category) with different tones and shown consecutively within one audiovisual video (see section 2.3. for detailed description). One video therefore consisted of the first image shown for the duration of the first sound (400 ms), a 60 ms break, and of the second image shown for the duration of the second sound (400 ms), forming a one audiovisual video of the length of 860 ms.

were manufactured into a Sin waveform with 0.8 intensity using Audacity (version 2.1.2–1, Carnegie Mellon University) [43].

Next, short video clips were prepared by pseudo-randomly pairing up two different audiovisual stimuli with keeping the stimulus category consistent. The length of one video (i.e., one trial) consisted of the first tone (presented together with the first image for 400 ms), a 60 ms break in between, and of the second tone (presented with the second image for 400 ms). Thus, an 860 ms audiovisual video was formed in which the visual element varied in one of the three dimensions, and both of the images used were paired with a different tone. Examples of such pairs are shown in Fig 1. Consequently, these videos were either congruent or incongruent according to the following, category-specific rules: "The higher the spatial elevation, the higher the tone", "the more dots presented, the higher the tone", and "the higher the number presented, the higher the tone". A total of 180 videos were prepared; 30 congruent and 30 incongruent for all three stimuli displays type conditions (spatial elevation, symbolic magnitude, and non-symbolic magnitude).

The stimuli were presented using Presentation® software (Version 18.0, Neurobehavioral Systems), on a laptop with a screen size of 13.3 inches, running windows 10 as OS [44].

## Procedure and design

In the present study, a 2x2x3 mixed factorial design was implemented with group (musical training, no musical training) as the between subject factor, and congruency (congruent, incongruent), and stimulus dimension (spatial elevation, symbolic magnitude, and non-symbolic magnitude) as the within subject factors. The dependent variables were response latencies (in milliseconds) and accuracy (number of correct responses).

The participants were seated comfortably in a quiet well-lit room approximately 60 cm away from the screen, with their right hand on keyboard. All participants received written instructions, and provided informed consent in a written form prior to the experiment. The auditory stimuli were provided via Shike QHP-660 headphones. First, example trials of both congruent and incongruent stimuli from each of the categories were provided to the participants (6 example trials in total). It was then verbally confirmed that the instructions were understood. Next, the experiment began. The experimental procedure is shown in Fig 2.

Each trial consisted of a single video clip (a single panel on Fig 2). The sequence of events in each trial was as follows: the first audiovisual stimulus was presented in the middle of the screen for 400 ms, followed by a short break of 60 ms before the second audiovisual stimulus (400 ms). Each of the audiovisual stimuli consisted of a visual picture (see Fig 1) associated with an auditory stimulus that varied in pitch. The onset time for the auditory stimuli was synchronized with the onset of the visual images. After presenting both audiovisual stimuli (i.e., after presenting a single trial consisting of a pair of audiovisual stimuli), congruency was estimated; the pair of audiovisual stimuli allowed the participants to estimate congruency between the stimuli based on the explicitly learned abstract rules. These rules were "The higher the spatial elevation, the higher the tone", "the more dots presented, the higher the tone", and "the higher the number presented, the higher the tone", depending on the category. The participant were instructed to respond as quickly and accurate as possible with their right hand only by pressing 'K' on the keyboard when the corresponding rules were followed, and pressing the button 'L' when the rules were not followed. After the response, a 1000 ms blank screen appeared before the next trial. The researcher further instructed the participants verbally, and made sure the instructions were understood properly.

There were two experimental blocks, and each block consisted of 180 trials. In each block, there were three audiovisual stimulus categories with 60 trials in each (30 congruent and 30

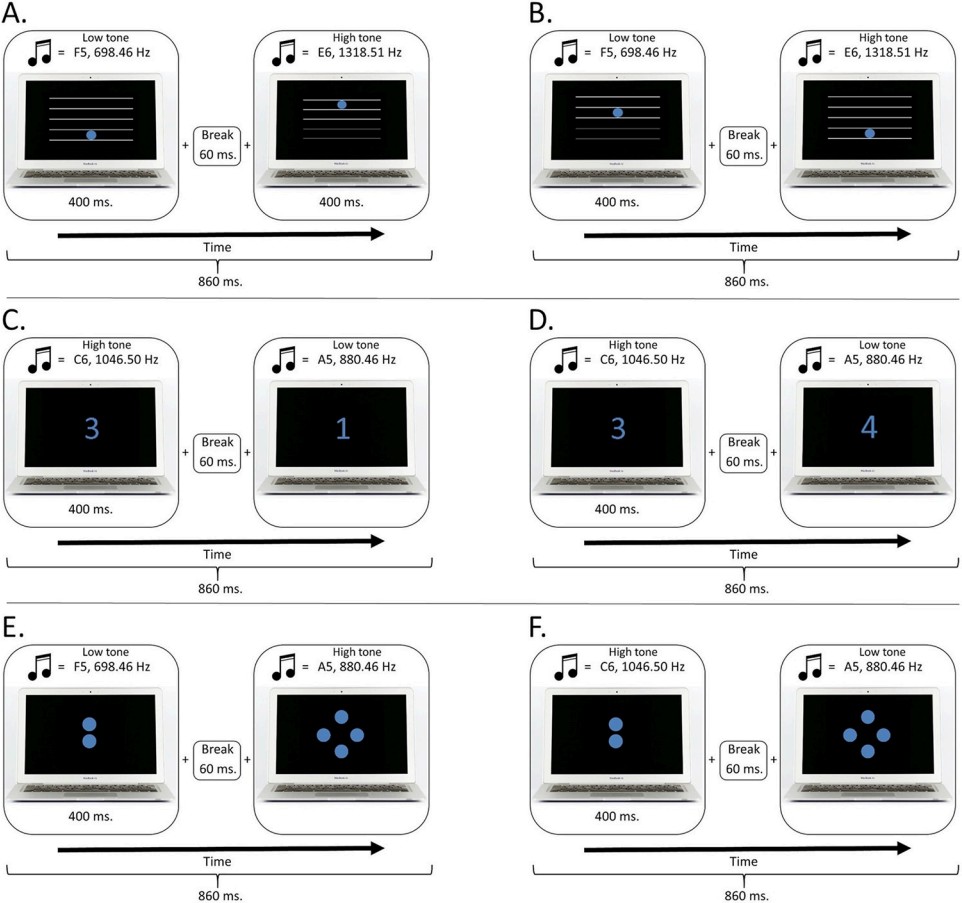

**Fig 2. Examples of the experimental procedure in both congruent and incongruent conditions for each of the tree audiovisual stimulus category.** Panels **A–B** illustrate congruent and incongruent trials in the pitch-elevation category, respectively. Each panel/trial consisted of a single video with the total length of 860 ms. In this video clip, the participant saw two visual stimuli each associated with a sound on a particular pitch. The pitch varied in frequency and the visual stimulus varied in elevation. Congruency was estimated according to an explicitly learned rule, "The higher the spatial elevation, the higher the tone". Panels **C–D** illustrate congruent and incongruent trials in the symbolic magnitude-pitch category, respectively. Here, the stimulus varied in value of the number shown and congruency was estimated according to the explicitly learned rule, "the higher the number presented, the higher the tone". Panels **E–F** illustrate congruent and incongruent trials in the non-symbolic magnitude-pitch category, respectively. Here, the stimulus varied in the number of circles shown and congruency was estimated according to the explicitly learned rule, "the more dots presented, the higher the tone". Note that in each category congruency/incongruency could be induced either with the visual or with the auditory part of the stimuli.

incongruent). The order of the trials was pseudo-randomized across the participants so that there were not consecutive trials having the same stimulus type. This randomization process also enabled the elimination of any potential bias caused by varying intervals between the auditory tones and the visual stimuli.

## Analysis

The required number of participants was estimated by conducting a statistical power analysis based on the raw data of Paraskevopoulos et al. [24], which investigated differences in cortical responses between musicians and non-musicians, utilizing a similar method to the one in the present study. The effect size (*d*) in that study was 1.76, considered as large using Cohen's [45] criteria. With an alpha of .05, both the power table and the sample size table were calculated

using fpower (version 1.2–1, [46]) suggesting power >.99 with $N = 10$ for each experimental group.

We applied a standard in which RTs below 100 ms and above 1000 ms, as well as two standard deviations from each condition's individual mean were considered as outliers and thus eliminated from the data [47]. In addition, trials in which the participant did not provide a response, or made a mistake were excluded from the reaction time analysis. As our reaction times were measured from the onset of the first picture within the video clip, response latencies below 560 ms (first picture + break + 100 ms) and above 1460 ms (first picture + break + 1000 ms) were eliminated. Participants who did not reach a level of 50% accepted trials per condition were disregarded from the analyses. This way we ensured at least 30 trials per condition for each participant in the subsequent analysis, and were left with a sample of 44 participants (25 musicians; 24 females). Maximum number of mistakes made per category was 28.

For the accuracy of following the rules, the discriminability index $d$ prime was calculated. For the calculations of the d-prime, hits were defined as congruent stimuli correctly identified as congruent, misses as congruent identified as incongruent, false alarms as incongruent stimuli identified as congruent, and correct rejections as incongruent stimuli identified as incongruent. As our main interest was the potential difference in accuracy between musicians and non-musicians and as several participants made zero mistakes in some of the stimulus categories, instead of calculating the d-prime for each audiovisual category individually, following the suggestions of Stanislaw & Todorov [48], we combined the data from the three stimulus categories before calculating the hit and false-alarm rates. For the further investigation of audiovisual category-wise accuracy, we used the raw number of mistakes. As the data were not normally distributed, a Mann-Whitney U test was performed. A significant effect was found and the effect size ($r$) was calculated. As the number of mistakes reached 7.5%, the mistakes across the visual stimulus categories were analysed using the Friedman test, and between the two congruency categories with Wilcoxon Signed-Rank test.

For reaction times, a mixed 2x2x3 ANOVA was performed with musical training as the between-participants factor, and congruency and stimulus categories as within-participant factors. Bonferroni corrected estimated marginal means were calculated for the main effects when appropriate. Furthermore, as the experimental design subjected the results to a possible speed-accuracy trade-off (SAT), we calculated Linear Integrated Speed-Accuracy Scores (LISAS) over all the categories [40–42]. LISAS were calculated by transforming the measurement scores to equal scales:

$$LISAS_{ij} = \begin{cases} RT_{ij} \ if \ PE_{ij} = 0 \\ RT_{ij} + PE_{ij} \times \dfrac{S_{RT_j}}{S_{PE_j}} \ otherwise \end{cases}$$

where $S_{RT}$ and $S_{PE}$ are the standard deviations of the participants' reaction times and proportion of errors, respectively. A 2x2x3 ANOVA was then performed on these transformed scores, with corresponding Bonferroni post-hoc comparisons.

## Results

### Reaction times

The results of the ANOVA (see Table 1) showed statistically significant main effects for congruency ($F(1,42) = 80.78$, $p < .001$, $\eta p2 = 0.658$), and for audiovisual stimulus category ($F(2,84) = 34.46$, $p < .001$, $\eta p2 = 0.451$), but not for musical training group ($p = .902$). That is, overall response times were faster for congruent trials ($M = 1452.76$ ms) than for incongruent

**Table 1. Audiovisual category-wise mean response latencies (SD) in milliseconds for musicians and non-musicians in spatial elevation, non-symbolic magnitude, and symbolic magnitude categories, in both, congruent and incongruent conditions.** At bottom, overall means (SE) for the audiovisual categories, musicians and non-musicians, and for congruent and incongruent stimuli.

| Musical Training | Congruency | Spatial Elev. | Symbolic Mag. | Non-symbolic Mag. |
|---|---|---|---|---|
| Musician | Cong. | 1389.90 (179.79) | 1459.86 (221.87) | 1480.34 (235.78) |
| | Incong. | 1530.09 (258.68) | 1634.61 (293.08) | 1656.11 (285.68) |
| Non-musician | Cong. | 1422.07 (216.33) | 1506.05 (245.72) | 1458.35 (213.98) |
| | Incong. | 1511.94 (230.49) | 1586.34 (236.34) | 1613.94 (261.20) |
| | Overall mean (SE) | 1463.50 (32.61) | 1546.72 (37.21) | 1552.19 (37.10) |
| Mean (SE) musician | 1525.15 (46.03) | | | |
| Mean (SE) non-musician | 1516.45 (52.81) | | | |
| Mean (SE) congruent | 1452.76 (32.26) | | | |
| Mean (SE) incongruent | 1588.84 (39.09) | | | |

trials ($M$ = 1588.84 ms) but did not significantly differ between musicians and non-musicians. Bonferroni corrected estimated marginal means showed that response times were significantly faster for the spatial elevation category ($M$ = 1463.50 ms, $SE$ = 32.61) than for the symbolic (1546.72 ms; $p$ < .001, $SE$ = 37.21) and non-symbolic magnitude (1552.19 ms, $p$ < .001, $SE$ = 37.10) categories, which did not significantly differ from each other ($p$ = 1.00). None of the interactions reached statistical significance (all $p$s > 0.077).

## Accuracy

The assumption of normality was found to be violated for the $d$ primes: in a Shapiro-Wilk, $W$ (44) = 0.94, $p$ = 0.018. Therefore, the $d$ prime data was analysed with non-parametric tests.

Both musicians and non-musicians scored significantly higher than the chance level indicating that both groups made more correct responses than could have been expected by chance alone. In Wilcoxon Signed-Ranks tests, musicians: $Mdn$ = 4.44, $Z$ = -4.37, $p$ < .001; non-musicians: $Mdn$ = 3.30, $Z$ = -3.82, $p$ < .001.

A Mann-Whitney U test was conducted for comparison of the two groups (musicians and non-musicians). The analysis revealed a statistically significant difference in the discriminability index. That is, musicians made significantly fewer mistakes than non-musicians (U = 85, $p$ < .001, $d$ = 1.299, large effect). Fig 3 shows the difference in the mean d-prime between musicians and non-musicians discriminating between congruent and incongruent stimuli.

Moreover, while musicians and non-musicians did not significantly differ in the number of mistakes in congruent categories (in a Mann-Whitney U test, $p$ = 0.113), in incongruent trials musicians ($M$ = 12.64) made significantly fewer mistakes than non-musicians ($M$ = 47; U = 38, $p$ < .001, $d$ = 2.032, large effect).

In analysis of number of mistakes in each audiovisual category individually, musicians made statistically significantly fewer mistakes than non-musicians with congruent non-symbolic stimuli (in a Mann-Whitney U test, U = 152.5, $p$ = 0.038, $d$ = 0.637, medium effect), incongruent elevation (U = 42, $p$ < .001, $d$ = 1.951, large effect), incongruent symbolic magnitude (U = 47, $p$ < .001, $d$ = 1.857, large effect), and incongruent non-symbolic magnitude (U = 37.5, $p$ < .001, $d$ = 2.042, large effect).

A Wilcoxon Signed-Rank test revealed a statistically significant, large difference between the overall number of mistakes made in congruent ($M$ = 15.5, $SD$ = 24.44) and incongruent ($M$ = 27.48, $SD$ = 24.74; $Z$ = 3.30, $p$ = 0.001, $r$ = 0.50) trials, indicating that the participants were more prone to make mistakes in incongruent than congruent categories.

The analysis of the overall number of mistakes made between elevation ($M$ = 13.14, $SD$ = 13.89), symbolic magnitude ($M$ = 14.66, $SD$ = 14.92), and non-symbolic magnitude

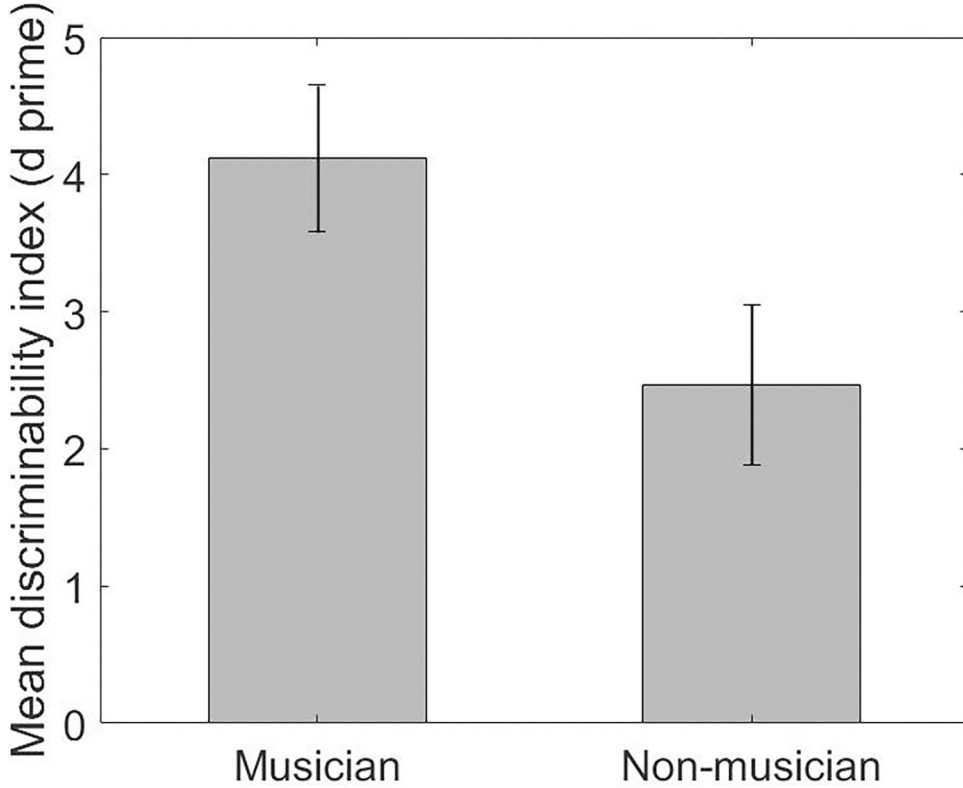

**Fig 3. Mean _d_ primes for musicians and non-musicians discriminating between congruent and incongruent trials.** Higher scores for musicians reflect higher accuracy. Error bars indicate the 95% confidence intervals. The between group difference is significant at $p < .001$.

($M$ = 15.18, $SD$ = 15.10) indicated a statistically significant difference in the number of made mistakes: in a Friedman test $x^2(2)$ = 13.87, $p$ = 0.001. Post hoc comparisons (Bonferroni corrected α = 0.017) revealed that elevation category differed significantly from non-symbolic magnitude ($Z$ = -3.37, $p$ = 0.001, $r$ = .51). Comparisons between elevation and symbolic magnitude and between symbolic magnitude and non-symbolic magnitude did not reach statistical significance ($p$s > 0.043 and 0.656, respectively).

### Linear Integrated Speed-Accuracy Scores (LISAS)

The mean scores (SD) for LISAS are shown in Table 2. The results of a mixed ANOVA indicated statistically significant main effects for congruency ($F(1,42)$ = 80.09, $p < .001$, $\eta p^2$ = .656), and for stimulus category ($F(2,84)$ = 35.99, $p < .001$, $\eta p^2$ = .461), but not for musical training group ($p$ = .79). In particular, LISAS scores were smaller for congruent ($M$ = 1457.49)

**Table 2. Mean linear integrated speed-accuracy scores (SD) and means (SD) for spatial elevation, non-symbolic magnitude, and symbolic magnitude categories, in both, congruent and incongruent conditions.**

| Musical Training | Congruency | Spatial Elev. | Symbolic Mag. | Non-symbolic Mag. |
|---|---|---|---|---|
| Musician | Cong. | 1394.54 (180.45) | 1470.86 (224.30) | 1488.05 (238.47) |
|  | Incong. | 1539.39 (258.80) | 1656.70 (297.02) | 1678.90 (288.83) |
| Non-musician | Cong. | 1423.44 (216.81) | 1507.82 (255.79) | 1460.24 (214.82) |
|  | Incong. | 1515.60 (231.27) | 1589.95 (237.55) | 1618.11 (261.68) |

than for incongruent trials ($M$ = 1599.77). The Bonferroni corrected estimated marginal means indicated that elevation category ($M$ = 1468.24) differed from both symbolic ($M$ = 1556.33, $p < .001$) and non-symbolic magnitudes ($M$ = 1561.32, $p < .001$), while there was no statistically significant difference observed between the two magnitude categories ($p = 1.00$).

In addition, we observed a statistically significant interaction between congruency and category of the stimuli ($F(2,84)$ = 3.34, $p = 0.04$, $\eta p^2$ = .074). Bonferroni corrected estimated marginal means showed that with both congruent and incongruent stimuli, elevation category was significantly different from symbolic and non-symbolic magnitudes ($ps < .001$); in congruent condition the difference was larger between elevation and symbolic magnitudes, while in the incongruent condition the difference was larger between elevation and non-symbolic magnitudes. All other interactions were statistically non-significant (all $ps > .0.54$).

## Discussion

Our main objective was to explore the association between long-term musical training and bimodal sensory integration. In a decision reaction-time task, we presented crossmodally correspondent audiovisual stimuli to musicians and non-musicians and measured both accuracy and response latencies. Our main result indicated a large, significant advantage for musicians, relative to non-musicians, in accuracy, but not in reaction times. Importantly, this observed advantage was present not only with pitch-elevation category, but extended also over two magnitude categories. Moreover, this advantage in accuracy was especially prominent with incongruent audiovisual stimulus categories; musicians had an advantage in all three incongruent stimulus categories, in addition to congruent non-symbolic magnitudes.

As expected, we observed significant main effects of congruency with all measures (response times, $d$ primes, number of mistakes, and linear integrated speed-accuracy scores; LISAS), suggesting audiovisual integration for all three stimulus categories, both for musicians as well as for non-musicians. The reaction times (RTs) were overall faster for the spatial elevation category relative to the other two categories, with the number of mistakes made with elevation being significantly smaller than with non-symbolic magnitude. This speed advantage with the spatial elevation category could reflect more extensive experience and/or innateness of naturally occurring statistics between pitch and elevation [12, 20, 49]. These results were further confirmed with LISAS, combining response latencies and the number of errors [42]. However, we did not observe the hypothesized difference in reaction times between musicians and non-musicians. The present pattern of findings is best explained in terms of an interaction between explicit and implicit processing and suggest a way by which the benefits of musical training on performance may generalize beyond the musical context.

Hence, contrary to what was hypothesized, no advantage in response latencies for musicians was established in the present study, whereas a significant advantage in accuracy was found. These results do not agree with previous studies in which a processing speed advantage for musicians has been observed, specifically with multisensory stimuli when compared to non-musicians, in stimulus detection [30] and in perceptual illusory tasks [31]. One possible explanation for the discrepancy in the results between previous studies and the present study may be in task-specific cognitive processing demands. First, while in previous similar studies participants were instructed to attend and make a single judgment (e.g., detect an auditory stimulus, tactile stimulus, or an audio-tactile stimulus), in the present study, participants were asked to switch between three different judgments–based on explicitly learned abstract rules–within the same block (pitch + spatial elevation or magnitudes). Thus, we cannot rule out the possibility that switching cost effects could have influenced performance in the task.

However, von Bastian & Druey [50], using a latent factor model, concluded that, from all types of switching (judgment, dimension, stimulus, mapping and response), response mapping shifting is the most relevant (the only one contributing directly) to switching cost. In our study, the response mapping remained the same across the judgment conditions, and thus, we believe that it is unlikely that switching cost effects could explain the lack of differences between musicians and non-musicians on response latencies. Given that response latency is the most sensitive measure of switching cost, and that the group effects were not significant with response latencies, we also believe that a potential musician advantage on judgment switching ability does not account for the group differences (better performance) found in accuracy data.

Second, in both, Laundry & Champoux [30] and in Bidelman [31], the task and the manipulations operated exclusively at the perceptual level. It remains possible that such stimulus detection or perceptual illusory effects do not require higher cognitive processing, whereas a task such as in the present study–where participants are required to be simultaneously cognizant of both modalities (pitch + spatial elevation or magnitudes) for a relatively long time, after which a comparison is made according to a newly learned abstract rule–does. Accordingly, short-term improvements, induced by short-term training on temporal binding windows have been found to be drastically similar despite alterations in task structure [51]. Crucially, this has been interpreted as to indicate that the effects induced by expertise reflect changes in perceptual rather than in higher cognitive systems.

We observed a significant advantage for musicians in accuracy, but not in reaction times, that extended over the magnitude categories. Importantly, previous research has suggested that comparative magnitude judgements require "higher" cognitive processing that encompasses more widely distributed cortical areas, including the parietal network [21, 39, 52]. On the other hand, response speed advantages for musicians have been observed with potentially cognitively less demanding tasks [30, 31]. Following this line of thought one may speculate that the present pattern of findings can be explained in terms of an interaction between top-down, higher order and pre-attentive processing–to the extent that distinguishing higher order and pre-attentive processing behaviourally is possible. That said, this potential interaction could be further broken down in future studies by, for example, including multiple tasks varying in cognitive demand and by analysing not only behavioural, but neuroimaging measures as well.

Furthermore, the observed lack of group effects with latency data could be due to differences in strategies and sound processing differences between musicians and non-musicians [53–55], which affected reaction times for the former. For instance, Chartrand & Belin [56] investigated timbre processing in a discrimination task, and found better performance for musicians, relative to non-musicians, in the discriminability index, but slower overall reaction times for musicians. They concluded that musicians might process sounds on a deeper level needing more time to encode the stimuli, which may result in a different strategic approach to the task (see also [57]). Therefore, it is possible that the observed reaction times for musicians in comparison to non-musicians reflect a trade-off between accuracy and speed. Motivated by this, we calculated LISAS scores to account simultaneously for both accuracy and response speed (see Methods). Crucially, the group factor did not yield statistically significant effects, and thus, further research is needed on the dissociation between latency and accuracy measures when investigating the relationship between musical training and multimodal sensory integration. These results furthermore support the association between implicit processing–as reflected on RTs–and explicit processing–as reflected on accuracy of judgements.

Interestingly, musicians–in comparison to non-musicians–had greater discriminability (*d* prime), and they made significantly fewer mistakes in all the incongruent audiovisual stimulus

categories, whereas they had an advantage only in one congruent category, non-symbolic magnitudes, and only with one measure (raw number of mistakes). As far as we are aware, there are no previous studies indicating whether the musicians' advantage in multisensory processing is more predominantly grounded on the processing of congruent or incongruent stimuli. Studies in the field typically use *d* prime as an index of accuracy, which identifies the signal detection but cannot attribute the difference on either of the two conditions. Nonetheless, congruent processing relies mostly on predictive coding principles [58]. As such, the audiovisual stimuli in the congruent condition does not violate the underlying predictions, and hence, engages predominantly top-down mechanisms, which are not explicitly trained in musicians. As the cross-modal correspondences seem to be innate [12, 20, 49, 59–61], we can expect to see similar behaviour reflecting congruent processing in both, musicians and non-musicians. On the other hand, stimuli in the incongruent condition violates the predictions of congruency engaging bottom-up processing, which benefits from the perceptual learning effects that musicians gain throughout their training, advancing the perceptual processing of the multisensory stimuli. In other words, it is possible that due to more refined multisensory integration and sharpened perception in relation to such stimuli, musicians can better identify the violated prediction of congruency in comparison to non-musicians. Such an advantage has previously been observed in relation to the temporal window of integration, where musicians show enhanced ability to detect audiovisual asynchrony [7]. These results conjointly with the results with RTs support the possible interaction between top-down and bottom-up processes.

Although we had–based on previous findings–reasonable grounds for hypothesising an advantage in overall performance induced by musical training, it could be that if indeed musical training enhances multisensory integration, it may have a measurable effect not in overall performance but in the size of the congruency effect. It would be worthwhile for a future study to explore whether such a larger congruency effect exists in musicians when compared to non-musicians. This is particularly true in the light of the observed lack of advantage in overall performance when measured with RT. We leave further investigation in this direction to future studies.

In addition, and as expected, we found significant main effects of congruency with all measures (response times, *d* primes, number of mistakes, LISAS), suggesting audiovisual integration for all three stimulus categories. With regard to stimulus category, reaction times were faster for multisensory stimuli in which the visual dimension varied in spatial elevation, relative to the two magnitude categories. This observation was further complemented by the accuracy data; fewer mistakes were made with pitch-elevation in relation to non-symbolic magnitude category. Similarly, with LISAS, a main effect of visual stimulus category was found with advantage for spatial elevation category, suggesting that even when accuracy scores are taken into account, the trials in the spatial elevation category were processed more efficiently.

Taken together, this suggests a significant difference in the processing of the three types of multisensory stimuli. These faster responses in pitch-elevation category may indicate an overlearned and partly automated association between pitch and elevation in contrast to correspondences based more solely on newly learned abstract rules (the magnitude categories). Indeed, several previous studies have established pitch-elevation correspondence [14, 15, 20, 62, 63], relating it to naturally occurring statistics [49], and suggesting its innateness [59–61]. Interestingly, in relation to magnitudes–assuming no naturally occurring correspondence between pitch and symbolic magnitude exists (however see [36] and [64])–both musicians and non-musicians were able to internalize and automatize the abstract rule within a relatively short period, as both groups performed significantly better than what could be expected by chance.

We also observed that more mistakes were made overall in incongruent trials than in congruent trials. These results are consistent with previous findings on congruency effects with

crossmodal correspondences (see, [11, 13] for reviews) and support the notion that the benefits of musical training on performance may generalize beyond the musical context. This advantage was reflected also on congruency based on newly learned abstract rules in otherwise unrelated stimuli (pitch-magnitude pairs).

Finally, we also observed small, statistically significant interaction between congruency and stimulus category with LISAS. However, the estimated effect size was relatively small, and it has been suggested that with high powered studies, as the present one, p-values of 0.04 are more likely under the null-hypothesis (for a discussion, see [65]).

It is worth explicitly noting that similarly to most of the studies in this field, the methodology adopted here limits the conclusions we can draw such that a causal relationship cannot be established between musical training and advantages in multisensory integration; we can, at best, only show a relationship between the two factors. This is particularly true as the general cognitive abilities of the participants were not directly controlled, and hence other factors may play a role in explaining the observed differences between the two groups. Future studies should collect more extensive background knowledge regarding participants' cognitive abilities (e.g. level of education in general) and introduce control tasks to have a stronger basis for suggesting causality.

Another potential limitation of this study is that the musical instruments in our sample were considerably heterogeneous with most musicians reporting multiple instruments as their main skill. It is plausible that different instruments elicit different effects on multisensory integration, or more so, on some modalities over others. Therefore, future studies should investigate the specific effects of training on particular instruments.

## Conclusions

This study investigated the association between musical expertise and processing of audiovisual crossmodal correspondences varying in three visual stimulus dimensions (elevation, symbolic magnitude, non-symbolic magnitude) and in pitch. Congruency was assessed based on explicitly learned abstract rules in pitch-elevation and in otherwise unrelated pitch-magnitude stimuli. We provided novel evidence supporting an interaction between implicit processing–as reflected on RTs–and explicit processing–as reflected on accuracy of judgements. In particular, we observed a large effect on accuracy supporting the hypothesis that long-term musical training induced neuroplasticity effects on audiovisual integration can be generalized outside of musical stimuli. This advantage was reflected on congruency based on newly learned abstract rules in otherwise unrelated stimuli (pitch-magnitude pairs), suggesting an advantage on processes requiring higher order cognitive functions. Moreover, musicians performed better than non-musicians predominantly in the incongruent condition. These results are discussed in relation to the predictive coding framework. However, this advantage was not reflected on reaction time measures, or on scores accounting for speed-accuracy trade-offs. Thus, our study support the notion that accuracy and latency measures reflect different cognitive processes. Further research is needed to understand why musical training effects may affect differentially these performance measures, and what this means for the corresponding cognitive processes.

## Author Contributions

**Conceptualization:** Riku Ihalainen, Evangelos Paraskevopoulos.

**Data curation:** Riku Ihalainen, Georgios Kotsaridis.

**Formal analysis:** Riku Ihalainen.

**Investigation:** Riku Ihalainen, Georgios Kotsaridis.

**Methodology:** Riku Ihalainen, Evangelos Paraskevopoulos.

**Resources:** Ana B. Vivas, Evangelos Paraskevopoulos.

**Supervision:** Evangelos Paraskevopoulos.

**Visualization:** Riku Ihalainen.

**Writing – original draft:** Riku Ihalainen, Ana B. Vivas, Evangelos Paraskevopoulos.

**Writing – review & editing:** Riku Ihalainen, Ana B. Vivas, Evangelos Paraskevopoulos.

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
