## [Decision Letter · Decision Letter 0]

5 Apr 2022

PONE-D-22-01297 The effect of musical training on the processing of audiovisual correspondences: Evidence from a reaction time taskPLOS ONE

Dear Dr. Ihalainen,

Thank you for submitting your manuscript to PLOS ONE. After careful consideration, we feel that it has merit but does not fully meet PLOS ONE’s publication criteria as it currently stands. Therefore, we invite you to submit a revised version of the manuscript that addresses the points raised during the review process. The reviewers have provided careful and constructive comments which I feel can be addressed in a thorough revision. Please pay particular attention to clarifying the procedure and reporting the statistics correctly. 

We look forward to receiving your revised manuscript.

Kind regards,

Deborah Apthorp, Ph.D

Academic Editor

PLOS ONE

Journal Requirements:

Reviewers' comments:

Reviewer's Responses to Questions

**Comments to the Author**

1. Is the manuscript technically sound, and do the data support the conclusions?

Reviewer #1: Partly

Reviewer #2: No

2. Has the statistical analysis been performed appropriately and rigorously? 

Reviewer #1: No

Reviewer #2: Yes

3. Have the authors made all data underlying the findings in their manuscript fully available?

Reviewer #1: Yes

Reviewer #2: No

4. Is the manuscript presented in an intelligible fashion and written in standard English?

Reviewer #1: Yes

Reviewer #2: Yes

5. Review Comments to the Author

Reviewer #1: This paper investigates the effect of musicianship on three audiovisual crossmodal correspondences: pitch-elevation/symbolic & non-symbolic magnitude (PE, PSM, PNSM, respectively). Musicianship had no effect on reaction times; but musicians were more accurate than non-musicians overall, largely because they were more accurate on incongruent trials. Accuracy was highest on the PE task but there appear to be no differences between musicians and non-musicians on any task. Similarly, there is no group effect or group/task interaction for speed-accuracy scores. The authors attribute these results to top-down effects on pre-attentive processing.

While this is an interesting question, it’s quite hard to work out what participants were actually asked to do and, thus, how the question is being addressed. Each trial consists of two audiovisual pairs, let’s say 1 dot and 4 dots each accompanied by a different tone. What exactly makes a trial congruent or incongruent? If the tones are different in each pair, one has to be higher than the other. Is a congruent trial one in which the single dot is accompanied by a low tone and the 4 dots by a high tone, with an incongruent trial being 1-dot/high tone plus 4-dots/low tone? If so, what’s the point of the pairs? You could achieve (in)congruency with a single audiovisual stimulus, i.e. congruent = 1-dot/low tone, incongruent = 1-dot/high tone. Or do participants have to say whether the second pair breaks the rule?

It would be helpful to amend Fig 1 to indicate how the tones are paired with the visuals (e.g., insert text ‘high’/‘low’), and to show both congruent and incongruent examples.

PE, PSM and PNSM trials are randomly ordered within a block of 180 trials, so participants have to remember three different rules and switch between them at random. To what extent do the results simply reflect task demands and switching costs?

Incidentally, the paper constantly refers to ‘visual stimuli’ and ‘visual stimulus categories’ when they are, in fact, audiovisual. In any case, what the authors seem to mean by this is the three crossmodal correspondences/tasks, it would be clearer to refer to them as such.

There also seems to be some confusion over what constitutes congruency: better performance in a multisensory condition than a unisensory condition reflects integration, not congruency (p5); see also comments on the Discussion.

Results

3.1 Please report the t-tests for the post-hoc comparisons for the main effect of task (and the Bonferroni-corrected alpha). Was the group/task interaction not significant? It would be helpful to add an overall mean to the bottom of Table 1 so that the reader can connect back to the text.

3.2 What does Figure 2 show? Does it refer to performance against chance (p14) – in which it should show what value would represent chance and also show median values as in the text instead of changing to mean values – or does it reflect the Mann-Whitney test (p15)?

The legend suggests that it refers to “musicians and non-musicians discriminating between congruent and incongruent trials”. This would be more useful than either of the options above but, in that case, the data would change to number of mistakes and for each group should be broken down by trial type. This is the “large advantage for musicians” (p18/367) so let’s see that clearly.

The final paragraph of 3.2 describes the main effect of trial type and would be better placed after the paragraph reporting the main effect of group.

Were there any group differences on each task?

3.3 Please report the means, SDs, tests, p-values, and corrected alpha for the explanation of the task/trial type interaction.

Was the group/task interaction not significant?

Discussion

Paragraph 1 needs to make clear that the advantage for musicians is quite general, across all tasks, with no group/task interactions (I assume, see comments above).

The opening sentence of paragraph 2 is a bit misleading – the results show *overall* main effects of congruency (faster RTs, fewer mistakes, smaller LISAS for congruent compared to incongruent) but no congruency/task interactions are reported for RTs or accuracy: presumably these were not significant? There is such an interaction in the LISAS analysis but, instead of a difference between congruent and incongruent trials *within* a task (i.e., a congruency effect), this seems to reflect differences in the congruent/incongruent conditions *between* tasks (and only the magnitude tasks) which is not helpful. Just because overall RTs are faster for the PE task compared to the other two doesn’t mean there’s a congruency effect.

Overall, I feel the authors need to be clearer about how they’re defining key terms and make sure these reflect accepted definitions in the literature and then go from there.

Minor points

p6/138: after ‘sets’ insert ‘out’

p8: the first paragraph should be moved to the start of section 2.4 where it makes more sense; the title for section 2.1 then becomes just ‘Participants’.

p12/260: if there are 2 experimental blocks each with 180 trials then all stimuli are presented twice?

p15/324, 333, 334: please report the statistical test supporting the p-value.

p16/344-345: the p-values are redundant, they just reflect the main effect reported a few lines above.

p16/346-348: please report the tests underlying the p-values and the corrected alpha level.

p12/262: ‘trials’ not ‘trails’

p18/376, p19/400: ‘innateness’ would be better.

p19/391: I’m not sure I would refer to the RT data as ‘raw’ because they were considerably cleaned up – perhaps ‘absolute’ or just not qualify it at all.

p20/421: after ‘previous’ insert ‘studies’.

Figure 1: the labels for symbolic and non-symbolic magnitude need to be swapped so that they are under the correct task.

Reviewer #2: The current study examined the influences of musical training on crossmodal correspondences between vision and audition. Three rules of correspondences were tested: pitch-elevation, pitch-numerosity, and pitch-digit pairings. The results demonstrated the only effect involves musical training was that musicians responded less errors than non-musicians, especially in the incongruent trials. In addition, responses were faster and less errors in the congruent than in the incongruent condition, and faster for the pitch-elevation pairing than for the pitch-numerosity and pitch-digit pairings; similar effect was demonstrated when considering both response time and errors using the index of LISAS.

In general, the rationale and the design of the study is confusing, so it is hard for me to reach any clear conclusion. Here are my main concerns:

1. The first concern is the rationale of using accuracy and response time measures. To my knowledge, accuracy is more suitable than response time when probing early processing of stimuli with time-limited presentation (Norman & Bobrow, 1975, Cognitive Psychology; Santee & Egeth, 1982, JEP:HPP). In contrast to the authors’ arguments, response time measure often involves the accumulation process of decision making.

2. It is unclear how to separate different types of crossmodal correspondences at pre-attentive stage associated with sub-cortical structure versus higher-order cognitive process. Presumably, sub-cortical structures mention by the authors (superior colliculus) does not represent stimulus identity, and therefore it is not possible to reveal any crossmodal correspondences at this level of processing.

3. It is unclear why the three correspondences rules for participants were defined as “newly learned”—did the participants truly learn the rule, or they were merely instructed to responded in such ways?

More specifically the rule “The higher the spatial elevation, the higher the tone” is a natural correspondence and has been repeatedly reported in literature (reviewed in Discussion). However, the other two rules “the more dots presented, the higher the tone” and “the higher the number presented, the higher the tone” seem to be counter-intuitive to the vertical numerical line (Hung, Hung, Tzeng, & Wu, 2008, Cognition). It is therefore not surprising that the response time for the first rule was faster than the latter two rules.

4. The authors’ prediction is ambiguous: If musical training induces enhanced multisensory integration, should the prediction be larger congruency effect rather than overall better performance for musicians than non-musicians?

5. The experimental design is confusing and hard to follow:

(1) There were four types of stimuli in each visual and auditory stimulus domain. Would it be possible that some trials would be easier (such as using the tones F5 and E6) than other trials (such as using the tones A5 and C6)?

(2) There were two audiovisual stimulus pairs presented sequentially in each trial. Isn’t one pair of audiovisual stimuli sufficient for response?

(3) A figure of experimental procedure would be helpful.

(4) How were the hit and false alarm rates defined when calculating d prime?

6. In Figure 2, there should be 2x3x2 bars, corresponding to the experimental design.

7. Can the better performance (less errors) in musicians than non-musicians simply reflect a better motor control after musical training of instruments?

6. PLOS authors have the option to publish the peer review history of their article (what does this mean?). If published, this will include your full peer review and any attached files.

Reviewer #1: No

Reviewer #2: No

---

## [Author Response · Author response to Decision Letter 0]

4 Oct 2022

Response to Reviewers

Reviewer #1 comments to the authors

This paper investigates the effect of musicianship on three audiovisual crossmodal correspondences: pitch-elevation/symbolic & non-symbolic magnitude (PE, PSM, PNSM, respectively). Musicianship had no effect on reaction times; but musicians were more accurate than non-musicians overall, largely because they were more accurate on incongruent trials. Accuracy was highest on the PE task but there appear to be no differences between musicians and non-musicians on any task. Similarly, there is no group effect or group/task interaction for speed-accuracy scores. The authors attribute these results to top-down effects on pre-attentive processing.

While this is an interesting question, it’s quite hard to work out what participants were actually asked to do and, thus, how the question is being addressed. 

Response to reviewer: The authors would like to sincerely thank the reviewer for the feedback, and especially for highlighting the weaknesses and raising the questions on issues that were not clearly communicated or considered in the original manuscript. We find the feedback very constructive, and have taken the utmost care to address your concerns in the revised manuscript.

As for making it more clear what the participants were asked to do, we have added an additional figure in the manuscript (figure 2) that shows the experimental procedure in detail in congruent and incongruent conditions for each of the three audiovisual stimulus categories. With the figure we have added the following legend: “Fig 2. Examples of the experimental procedure in both congruent and incongruent conditions for each of the three audiovisual stimulus category. Panels A – B illustrate congruent and incongruent trials in the pitch-elevation category, respectively. Each panel/trial consisted of a single video with the total length of 860 ms. In this video clip the participant saw two visual stimuli each associated with a sound on a particular pitch. The pitch varied in frequency and the visual stimulus varied in elevation. Congruency was estimated according to an explicitly learned rule, “The higher the spatial elevation, the higher the tone”. Panels C – D illustrate congruent and incongruent trials in the symbolic magnitude-pitch category, respectively. Here, the visual stimulus varied in value of the number shown and congruency was estimated according to the explicitly learned rule, “the higher the number presented, the higher the tone”. Panels E – F illustrate congruent and incongruent trials in the nonsymbolic magnitude-pitch category, respectively. Here, the visual stimulus varied in the number of circles shown and congruency was estimated according to the explicitly learned rule, “the more dots presented, the higher the tone”. Note that in each category congruency/incongruency could be induced either with the visual stimuli or with the auditory stimuli.”

1. Each trial consists of two audiovisual pairs, let’s say 1 dot and 4 dots each accompanied by a different tone. What exactly makes a trial congruent or incongruent? If the tones are different in each pair, one has to be higher than the other. Is a congruent trial one in which the single dot is accompanied by a low tone and the 4 dots by a high tone, with an incongruent trial being 1-dot/high tone plus 4-dots/low tone? If so, what’s the point of the pairs? You could achieve (in)congruency with a single audiovisual stimulus, i.e. congruent = 1-dot/low tone, incongruent = 1-dot/high tone. Or do participants have to say whether the second pair breaks the rule?

Response to reviewer: We do acknowledge that the task the participants were asked to do could have been communicated more clearly. Here, congruency/incongruency was defined by reflecting whether the abstract rule was followed or not. The reviewer is correct in that a congruent trial is one in which e.g. a single dot is accompanied by a low tone followed by 4 dots accompanied by a high tone. An incongruent trial, on the other hand, could be one in which a single dot accompanied by a high tone is followed by 4 dots accompanied by a low tone. One of such sequence formed a single trial and consisted of a pair of audiovisual stimuli (one dot with a tone and 4 dots with a tone). After such a pair was shown to the participant, the participants were instructed to indicate whether the rules were followed or not. 

It is not clear how such a reflection of the congruency (e.g. whether the rules were followed or not) could be achieved with a single audiovisual stimulus (i.e. one dot with a low tone). We do feel that the confusion here is due to our poor communication of the task (and what is meant by ‘a pair of audiovisual stimuli’). Hence, we have made the following changes to the corresponding parts of the manuscript:

We have added additional figure (figure 2) showing the detailed experimental procedure in congruent and incongruent conditions for each of the tree audiovisual stimulus categories. With the figure, we have written a detailed legend explaining the figure (and hence the procedure) in what we believe to be clear terms (see also reply above). Moreover, we have edited the methods section to explain the procedure more clearly (lines 250-290).

Lines 250-290 now say the following: “Next, the experiment began. The experimental procedure is shown in Fig 2.

Enter figure 2 here

Fig 2. Examples of the experimental procedure in both congruent and incongruent conditions for each of the tree audiovisual stimulus category. Panels A – B illustrate congruent and incongruent trials in the pitch-elevation category, respectively. Each panel/trial consisted of a single video with the total length of 860 ms. In this video clip, the participant saw two visual stimuli each associated with a sound on a particular pitch. The pitch varied in frequency and the visual stimulus varied in elevation. Congruency was estimated according to an explicitly learned rule, “The higher the spatial elevation, the higher the tone”. Panels C – D illustrate congruent and incongruent trials in the symbolic magnitude-pitch category, respectively. Here, the stimulus varied in value of the number shown and congruency was estimated according to the explicitly learned rule, “the higher the number presented, the higher the tone”. Panels E – F illustrate congruent and incongruent trials in the non-symbolic magnitude-pitch category, respectively. Here, the stimulus varied in the number of circles shown and congruency was estimated according to the explicitly learned rule, “the more dots presented, the higher the tone”. Note that in each category congruency/incongruency could be induced either with the visual or with the auditory part of the stimuli.

Each trial consisted of a single video clip (a single panel on Fig 2). The sequence of events in each trial was as follows: the first audiovisual stimulus was presented in the middle of the screen for 400 ms, followed by a short break of 60 ms before the second audiovisual stimulus (400 ms). Each of the audiovisual stimuli consisted of a visual picture (see Fig 1) associated with an auditory stimulus that varied in pitch. The onset time for the auditory stimuli was synchronized with the onset of the visual images. After presenting both audiovisual stimuli (i.e. after presenting a single trial consisting of a pair of audiovisual stimuli), congruency was estimated; the pair of audiovisual stimuli allowed the participants to estimate congruency between the stimuli based on the explicitly learned abstract rules. These rules were “The higher the spatial elevation, the higher the tone”, “the more dots presented, the higher the tone”, and “the higher the number presented, the higher the tone”, depending on the category. The participant were instructed to respond as quickly and accurate as possible with their right hand only by pressing ‘K’ on the keyboard when the corresponding rules were followed, and pressing the button ‘L’ when the rules were not followed. After the response, a 1000 ms blank screen appeared before the next trial. The researcher further instructed the participants verbally, and made sure the instructions were understood properly.

There were two experimental blocks, and each block consisted of 180 trials. In each block, there were three audiovisual stimulus categories with 60 trials in each (30 congruent and 30 incongruent). The order of the trials was pseudo-randomized across the participants so that there were not consecutive trials having the same stimulus type. This randomization process also enabled the elimination of any potential bias caused by varying intervals between the auditory tones and the visual stimuli.”

2. It would be helpful to amend Fig 1 to indicate how the tones are paired with the visuals (e.g., insert text ‘high’/‘low’), and to show both congruent and incongruent examples.

Response to reviewer: Please, see the above response.

3. PE, PSM and PNSM trials are randomly ordered within a block of 180 trials, so participants have to remember three different rules and switch between them at random. To what extent do the results simply reflect task demands and switching costs?

Response to reviewer: We would like to thank the reviewer for this interesting suggestion. Indeed the task employed involved a “judgment” switching component, but please notice that the task did not require shifting of the response mapping, which remained the same across the judgment conditions. According to Von Bastian, & Druey (2017), who systematically investigated all types of switching (judgment, dimension, stimulus, mapping and response), the response mapping shifting is the one which is central and most relevant to switching ability and switching cost. In addition, and given that RTs is the most sensitive measure of switching cost, and that the main effect of group was not significant (and this factor did not interact with the other factors) in the response time analyses it is unlikely that a better switching ability in musicians could explain our results. 

We discuss the cognitive demand of the task and its possible effects on the results in lines 440-496. This section also includes a paragraph specifically discussion the possible effect of task switching and our reponse to it. 

4. Incidentally, the paper constantly refers to ‘visual stimuli’ and ‘visual stimulus categories’ when they are, in fact, audiovisual. In any case, what the authors seem to mean by this is the three crossmodal correspondences/tasks, it would be clearer to refer to them as such.

Response to reviewer: Thank you for this comment.. We edited the MS to increase clarity. For example, in the beginning of section 2.2. Stimuli, instead of talking about “three types of visual stimuli” we know refer to the visual and auditory parts of the audiovisual stimuli. Similarly, in the legend of figure 1, we refer to just stimulus, instead of “visual stimulus”. Similar changes have been made in the manuscript where necessary.

5. There also seems to be some confusion over what constitutes congruency: better performance in a multisensory condition than a unisensory condition reflects integration, not congruency (p5); see also comments on the Discussion.

Response to reviewer: We do acknowledge the confusion in the use of the terms here. This part has been edited to say (lines 105-109): “There is a relatively large number of reaction time studies done with non-musicians in the context of multimodal sensory integration. These investigations have consistently reported better performance with multimodal stimuli: faster detection of the target in the multimodal condition relative to unimodal condition (see for example Miller, 1991; for reviews see Parise & Spence, 2013; Spence, 2011).

RESULTS

6. 3.1 Please report the t-tests for the post-hoc comparisons for the main effect of task (and the Bonferroni-corrected alpha). Was the group/task interaction not significant? It would be helpful to add an overall mean to the bottom of Table 1 so that the reader can connect back to the text.

Response to reviewer: We have added more information on paragraph 3.1. We now explicitly state that the interactions were statistically non-significant and that the post hoc comparison was made using the estimated marginal means, and not with t-tests. The reported p-values for the estimated marginal means are bonferroni corrected.

We have also added the overall means in table 1, as per the suggestion.

7. 3.2 What does Figure 2 show? Does it refer to performance against chance (p14) – in which it should show what value would represent chance and also show median values as in the text instead of changing to mean values – or does it reflect the Mann-Whitney test (p15)?

The legend suggests that it refers to “musicians and non-musicians discriminating between congruent and incongruent trials”. This would be more useful than either of the options above but, in that case, the data would change to number of mistakes and for each group should be broken down by trial type. This is the “large advantage for musicians” (p18/367) so let’s see that clearly.

Response to reviewer: We acknowledge that indeed it was unclear to what figure 3 (figure 2 in previous version) is referring to. It shows the difference in the d-primes between musicians and non-musicians discriminating between congruent and incongruent stimuli. To make this clearer, we have added the following sentence on paragraph 3.2. Accuracy: Fig 3 shows the difference in the mean d-prime between musicians and non-musicians discriminating between congruent and incongruent stimuli.

 As for why the d-prime is calculated combining the audiovisual stimulus categories, as we responded to reviewer #2 (corrections number 6.), when recording the mistakes, we had several participants who made zero mistakes in some of the audiovisual categories, especially with congruent stimuli, hence causing issues in the d-prime calculations. As our main interest lied in the potential difference between musicians and non-musicians, rather than between the types of visual stimuli, to overcome the issue caused by participants with zero mistakes, we followed the suggestions of Stanislaw & Todorov (1999) and combined the data from several categories before calculating the hit and false-alarm rates (https://link.springer.com/content/pdf/10.3758/BF03207704.pdf). Consequently, when we investigated the accuracy between the visual categories further, we used the raw number of mistakes in the analyses.

We have added this information also in paragraph 2.4. Analysis which now states “As our main interest lied in the potential difference in accuracy between musicians and non-musicians and as several participants made zero mistakes in some of the stimulus categories, instead of calculating the d-prime for each audiovisual category individually, following the suggestions of Stanislaw & Todorov (1999), we combined the data from the three stimulus categories before calculating the hit and false-alarm rates. For the further investigation of audiovisual category-wise accuracy, we used the raw number of mistakes.”

8. The final paragraph of 3.2 describes the main effect of trial type and would be better placed after the paragraph reporting the main effect of group.

Were there any group differences on each task?

Response to reviewer: We have moved the paragraph below the results describing the difference in raw mistakes between musicians and non-musicians, as suggested. However, we kept the results with d-primes and raw mistakes separate.

We have also added a paragraph describing the difference between musicians and non-musicians in each of the audiovisual categories individually (lines 341-346). Correspondingly, we added these results also in the discussion section (lines 390-392 & 407-409).

9. 3.3 Please report the means, SDs, tests, p-values, and corrected alpha for the explanation of the task/trial type interaction.

Was the group/task interaction not significant?

Response to reviewer: For the interactions in reaction times, we have the results from the ANOVA with estimated marginal means, however, we feel reporting the estimated marginal means (with SDs, p-values, and corrected p-values) is not crucial, since the results cannot be separated from chance under the null hypothesis with a high enough precision. Rather, we have explicitly stated that the interactions did not reach statistical significance, and provided the level above which all the p-values lay.

DISCUSSION

10. Paragraph 1 needs to make clear that the advantage for musicians is quite general, across all tasks, with no group/task interactions (I assume, see comments above).

Response to reviewer: See our response above (correction #8).

11. The opening sentence of paragraph 2 is a bit misleading – the results show *overall* main effects of congruency (faster RTs, fewer mistakes, smaller LISAS for congruent compared to incongruent) but no congruency/task interactions are reported for RTs or accuracy: presumably these were not significant? There is such an interaction in the LISAS analysis but, instead of a difference between congruent and incongruent trials *within* a task (i.e., a congruency effect), this seems to reflect differences in the congruent/incongruent conditions *between* tasks (and only the magnitude tasks) which is not helpful. Just because overall RTs are faster for the PE task compared to the other two doesn’t mean there’s a congruency effect.

Response to reviewer: This is true: reading the sentence did give a misleading impression. What we meant was that when measured with RTs, d-primes, number of mistakes and LISAS, we found a congruency effect. We have corrected this sentence to state “Specifically, our results showed significant main effects of congruency with all measures (response times, d primes, number of mistakes, and linear integrated speed-accuracy scores), suggesting audiovisual integration for all three stimulus categories.”

Moreover, we have re-structured and largely re-written the entire discussion section to better reflect the comments and suggestions by both of the reviewers. Hence, the entire discussion section should be re-visited.

MINOR POINTS

12. p6/138: after ‘sets’ insert ‘out’ - corrected

p8: the first paragraph should be moved to the start of section 2.4 where it makes more sense; the title for section 2.1 then becomes just ‘Participants’. - corrected

p12/260: if there are 2 experimental blocks each with 180 trials then all stimuli are presented twice? – This was indeed a bit unclear. It now states “There were two experimental blocks, and each block consisted of 180 trials. In each block, there were three audiovisual stimulus categories with 60 trials in each (30 congruent and 30 incongruent).” (lines 255-257).

p15/324, 333, 334: please report the statistical test supporting the p-value. - corrected

p16/344-345: the p-values are redundant, they just reflect the main effect reported a few lines above. – removed

p16/346-348: please report the tests underlying the p-values and the corrected alpha level. - corrected

p12/262: ‘trials’ not ‘trails’ - corrected

p18/376, p19/400: ‘innateness’ would be better. - corrected

p19/391: I’m not sure I would refer to the RT data as ‘raw’ because they were considerably cleaned up – perhaps ‘absolute’ or just not qualify it at all. - corrected

p20/421: after ‘previous’ insert ‘studies’. - corrected

Figure 1: the labels for symbolic and non-symbolic magnitude need to be swapped so that they are under the correct task. - corrected

Reviewer #2 comments to the authors

The current study examined the influences of musical training on crossmodal correspondences between vision and audition. Three rules of correspondences were tested: pitch-elevation, pitch-numerosity, and pitch-digit pairings. The results demonstrated the only effect involves musical training was that musicians responded less errors than non-musicians, especially in the incongruent trials. In addition, responses were faster and less errors in the congruent than in the incongruent condition, and faster for the pitch-elevation pairing than for the pitch-numerosity and pitch-digit pairings; similar effect was demonstrated when considering both response time and errors using the index of LISAS.

In general, the rationale and the design of the study is confusing, so it is hard for me to reach any clear conclusion. Here are my main concerns:

Response to reviewer: The authors would like to sincerely thank the reviewer for the feedback. We find the feedback very constructive, especially the feedback concerning clarity and the rationale for parts of the manuscript. We have taken the utmost care to address your concerns in the revised manuscript.

1. The first concern is the rationale of using accuracy and response time measures. To my knowledge, accuracy is more suitable than response time when probing early processing of stimuli with time-limited presentation (Norman & Bobrow, 1975, Cognitive Psychology; Santee & Egeth, 1982, JEP:HPP). In contrast to the authors’ arguments, response time measure often involves the accumulation process of decision making.

Response to the reviewer: We wish to thank the reviewer for pointing this out, and we fully agree with the notion that reaction times and accuracy measures tap onto different underlying processes. As such, each type of response yields different information, with reaction time reflecting more on implicit processing and accuracy more on explicit processing. This is indeed what we argue in the manuscript, and provide as a justification for incorporating both of these two measures; we aim to capture these differences by using both measures.

We have edited the wording throughout the manuscript to better reflect this comment and removed the parts of text explicitly referring to ‘requirements of conscious processing’ (lines 99-110). Instead, we now highlight the point that the two measures reflect different cognitive processes (implicit/explicit), and hence, provide complementary information of the underlying processes. 

Moreover, we have re-structured and largely re-written the entire discussion section to better reflect the comments and suggestions by both of the reviewers. The discussion section now includes parts discussing the results in relation to these two measures and the underlying cognitive processing. Hence, we recommend that the entire discussion section should be re-visited.

2. It is unclear how to separate different types of crossmodal correspondences at pre-attentive stage associated with sub-cortical structure versus higher-order cognitive process. Presumably, sub-cortical structures mention by the authors (superior colliculus) does not represent stimulus identity, and therefore it is not possible to reveal any crossmodal correspondences at this level of processing.

Response to the reviewer: We agree with the reviewer that behaviourally the distinction between higher order cortical processing and lower level sub-cortical processing is difficult, in not directly impossible. However, this was not the aim of the manuscript. Rather, we in the corrected manuscript speculate an interplay between top-down, higher order and pre-attentive processing, and provide some suggestion on how future studies could potentially break down this interaction further.

That said, a great line of research has been done on multisensory correspondences (Bidelman, 2016; Laundry & Champoux, 2017; Miller, 1991; Paraskevopoulos et al., 2014; Paraskevopoulos et al., 2012; Parise & Spence, 2013; Spence, 2011) and previous research has already identified the role of sub-cortical structures, including but not limited to the superior colliculi in the processing of such stimuli (e.g. Stein & Meredith, 1993; for a review see Alais, Newell & Mamassian, 2010). The mentioning of these structures in the manuscript is hence grounded on a-priori knowledge provided by such relevant literature. 

Moreover, we have re-structured and largely re-written the entire discussion section to better reflect the comments and suggestions by both of the reviewers. Hence, the entire discussion section should be re-visited.

3. It is unclear why the three correspondences rules for participants were defined as “newly learned”—did the participants truly learn the rule, or they were merely instructed to responded in such ways?

More specifically the rule “The higher the spatial elevation, the higher the tone” is a natural correspondence and has been repeatedly reported in literature (reviewed in Discussion). However, the other two rules “the more dots presented, the higher the tone” and “the higher the number presented, the higher the tone” seem to be counter-intuitive to the vertical numerical line (Hung, Hung, Tzeng, & Wu, 2008, Cognition). It is therefore not surprising that the response time for the first rule was faster than the latter two rules.

Response to the reviewer: We fully agree that the advantageous effect of pitch-elevation over pitch-symbolic magnitude and pitch-non-symbolic magnitude in and of itself is not surprising. Indeed, in the manuscript we do not claim that it is, and instead, we relatively shortly discuss it in relation to previous papers suggesting its innateness and more extensive experience in comparison to the two other types of audiovisual stimuli used in the experiment (the two magnitude categories). Indeed, precisely because of pitch-elevation has been suggested to be a naturally occurring statistic with which people in general – and not only musicians – have more extensive experience, we added the two other audiovisual categories. With pich-magnitude and pitch-non-symbolic magnitude – as we discuss in the paper – such a natural linkage is not that clear (although see the manuscript discussion section lines 544 – 558). 

Hence, our task also requires the participants to estimate congruence based on explicitly learned rule, binding other unrelated unisensory stimuli. Here, ‘newly/explicitly’ is particularly true with the two magnitude categories, as no natural linkage between magnitudes and pitch has been demonstrated, and by ‘learning’ we simply refer to the participants’ internalizing the rule, keeping it in mind, and discriminating the audiovisual stimuli accordingly. 

In addition to not having a clear, naturally occurring linkage with pitch, magnitude processing in and of itself has been argued to reflect “higher” cognitive processes (Dehaene, Piazza, Pinel & Cohen, 2003; Paraskevopoulos et al., 2014). Hence, adding the two magnitude categories in addition to pitch-elevation enabled us to investigate the hypothesised advantage for musicians in task varying in cognitive demand, which we hypothesised could reflect on the two measures used (RT and accuracy).

Lastly, it is not clear to us how the vertical mental numerical line (particularly with Chinese number words mentally aligned top-to-bottom) is particularly problematic here: the only change in spatial elevation was with the pitch-elevation category and all our visual stimuli were centred horizontally. Moreover, in the manuscript we do shortly discuss the SNARC and SMARC effects. Perhaps there is confusion relating to the word ‘higher’ in our abstract rules, which in this case refers to only a higher value represented, rather than to spatial elevation.

4. The authors’ prediction is ambiguous: If musical training induces enhanced multisensory integration, should the prediction be larger congruency effect rather than overall better performance for musicians than non-musicians?

Response to the reviewer: We wish to thank the reviewer for raising this interesting question. We have added parts into the discussion section discussing this point (lines 503 – 534). 

First, we point out that regarding the group difference between congruent and incongruent condition, as far as we know, there are no previous results to indicate whether the advantage grounds on the processing of the congruent or the incongruent stimulus, as the typical studies in the field use d prime as an idex – which cannot attribute the difference on either of the two conditions, but only infer signal detection. We then relate this observation and further discuss it in relation to predictive coding framework (Cross‐modal predictive processing depends on context rather than local contingencies - Dercksen - 2021 - Psychophysiology - Wiley Online Library). Particularly, we suggest that in the congruent condition, the underlying predictions are not violated, and as the correspondence is suggested to be innate, the predction should (behaviourally) work equally in the two groups (engagning predominantly top-down processing, which is not explicitly trained in musicians). On the other hand, the incongruent condition violates the prediction and hence, engages bottom-up processing routes. This route, in turn, takes advantage of the perceptual learning effects that musicians develop throughout their training, sharpening the perceptual processing of multisensory stimuli. In other words, due to the enhanced multisensory integration, and sharpened perception in relation to such stimuli, they can better identify the violation from the predicted congruency. We also see this effect in relation to the temporal window of integration, where musicians can more easily discriminate out-of-synch stimuli, because of their sharpened temporal window of integration curve (Multisensory integration of drumming actions: musical expertise affects perceived audiovisual asynchrony | SpringerLink).

We also point out that although our hypothesis was strongly grounded in previous studies, a worthwhile direction for future studies could be to not only look for the overall effects on congruency, but also to look at measures such as the size of the congruency effect. 

5. The experimental design is confusing and hard to follow:

(1) There were four types of stimuli in each visual and auditory stimulus domain. Would it be possible that some trials would be easier (such as using the tones F5 and E6) than other trials (such as using the tones A5 and C6)?

Response to the reviewer: It is indeed true that some of the auditory stimulus pairs had a larger interval between them than others. The same is true to some extent with the visual stimuli (for example with the elevation category in which the physical difference in spatial distance between the two stimuli varied between the trials). The pairs of stimuli were chosen pseudorandomly and both the auditory tones and the visual stimuli were adopted from previous studies by Paraskevopoulos (2012; 2014; 2015) to establish comparability between those studies and the present study – to the extent it is possible despite differences in methodology. 

Crucially however, the participants heard, saw, and responded to the same stimuli, but in pseudorandom order. Hence, any potential bias caused by varied difficulty across the trials was eliminated.

We have added this information explicitly to the manuscript (line 272-274): “This randomization process also enabled the elimination of any potential bias caused by varying intervals between the auditory tones and the visual stimuli.”

(2) There were two audiovisual stimulus pairs presented sequentially in each trial. Isn’t one pair of audiovisual stimuli sufficient for response?

Response to the reviewer: We do acknowledge that the task the participants were asked to do could have been communicated more clearly. Here, congruency/incongruency was defined by reflecting whether the abstract rule was followed or not. The reviewer is correct in that a pair of audiovisual stimuli is sufficient for estimating congruency, and this is indeed what we tried to communicate in the manuscript.

To make the procedure more clear, we have added additional figure (figure 2) showing the detailed experimental procedure in congruent and incongruent conditions for each of the tree audiovisual stimulus categories. With the figure, we have written a detailed legend explaining the figure (and hence the procedure) in what we believe to be clear terms (see also reply above). Moreover, we have edited the methods section to explain the procedure more clearly (lines 250-290).

Lines 250-290 now say the following: “Next, the experiment began. The experimental procedure is shown in Fig 2.

Enter figure 2 here

Fig 2. Examples of the experimental procedure in both congruent and incongruent conditions for each of the tree audiovisual stimulus category. Panels A – B illustrate congruent and incongruent trials in the pitch-elevation category, respectively. Each panel/trial consisted of a single video with the total length of 860 ms. In this video clip, the participant saw two visual stimuli each associated with a sound on a particular pitch. The pitch varied in frequency and the visual stimulus varied in elevation. Congruency was estimated according to an explicitly learned rule, “The higher the spatial elevation, the higher the tone”. Panels C – D illustrate congruent and incongruent trials in the symbolic magnitude-pitch category, respectively. Here, the stimulus varied in value of the number shown and congruency was estimated according to the explicitly learned rule, “the higher the number presented, the higher the tone”. Panels E – F illustrate congruent and incongruent trials in the non-symbolic magnitude-pitch category, respectively. Here, the stimulus varied in the number of circles shown and congruency was estimated according to the explicitly learned rule, “the more dots presented, the higher the tone”. Note that in each category congruency/incongruency could be induced either with the visual or with the auditory part of the stimuli.

Each trial consisted of a single video clip (a single panel on Fig 2). The sequence of events in each trial was as follows: the first audiovisual stimulus was presented in the middle of the screen for 400 ms, followed by a short break of 60 ms before the second audiovisual stimulus (400 ms). Each of the audiovisual stimuli consisted of a visual picture (see Fig 1) associated with an auditory stimulus that varied in pitch. The onset time for the auditory stimuli was synchronized with the onset of the visual images. After presenting both audiovisual stimuli (i.e. after presenting a single trial consisting of a pair of audiovisual stimuli), congruency was estimated; the pair of audiovisual stimuli allowed the participants to estimate congruency between the stimuli based on the explicitly learned abstract rules. These rules were “The higher the spatial elevation, the higher the tone”, “the more dots presented, the higher the tone”, and “the higher the number presented, the higher the tone”, depending on the category. The participant were instructed to respond as quickly and accurate as possible with their right hand only by pressing ‘K’ on the keyboard when the corresponding rules were followed, and pressing the button ‘L’ when the rules were not followed. After the response, a 1000 ms blank screen appeared before the next trial. The researcher further instructed the participants verbally, and made sure the instructions were understood properly.

There were two experimental blocks, and each block consisted of 180 trials. In each block, there were three audiovisual stimulus categories with 60 trials in each (30 congruent and 30 incongruent). The order of the trials was pseudo-randomized across the participants so that there were not consecutive trials having the same stimulus type. This randomization process also enabled the elimination of any potential bias caused by varying intervals between the auditory tones and the visual stimuli.”

(3) A figure of experimental procedure would be helpful.

Response to the reviewer: This again is an excellent suggestion. We have added a figure showing the experimental procedure (see fig.2 and the response above).

(4) How were the hit and false alarm rates defined when calculating d prime?

Response to the reviewer: We have added the following sentence on paragraph 2.4 Analysis (lines 300-304): For the calculations of the d-prime, hits were defined as congruent stimuli correctly identified as congruent, misses as congruent identified as incongruent, false alarms as incongruent stimuli identified as congruent, and correct rejections as incongruent stimuli identified as incongruent.

6. In Figure 2, there should be 2x3x2 bars, corresponding to the experimental design.

Response to the reviewer: When recording the mistakes, we had several participants who made zero mistakes in some of the audiovisual categories, especially with congruent stimuli, hence causing issues in the d-prime calculations. As our main interest lied in the potential difference between musicians and non-musicians, rather than between the types of visual stimuli, to overcome the issue caused by participants with zero mistakes, we followed the suggestions of Stanislaw & Todorov (1999) and combined the data from several categories before calculating the hit and false-alarm rates (https://link.springer.com/content/pdf/10.3758/BF03207704.pdf). Consequently, when we investigated the accuracy between the visual categories further, we used the raw number of mistakes in the analyses.

We have added this information also in the Analysis section which now states (lines 304-309)“As our main interest lied in the potential difference in accuracy between musicians and non-musicians and as several participants made zero mistakes in some of the stimulus categories, instead of calculating the d-prime for each audiovisual category individually, following the suggestions of Stanislaw & Todorov (1999), we combined the data from the three stimulus categories before calculating the hit and false-alarm rates. For the further investigation of audiovisual category-wise accuracy, we used the raw number of mistakes.”

7. Can the better performance (less errors) in musicians than non-musicians simply reflect a better motor control after musical training of instruments?

Response to the reviewer: It is our view that such a suggested advantage in motor control after persistent musical training would first and foremost affect reaction times and not so much accuracy. In other words, if the better performance was due to better motor control, we think we should have seen an advantage in the reaction times first, which we did not observe here. It is especially counterintuitive to think that an advantage would be seen in accuracy measures but not in reaction times, if it indeed were due to better motor control only. Hence, it is unlikely that the better performance was due to better motor control in musicians.

---

## [Decision Letter · Decision Letter 1]

15 Dec 2022

PONE-D-22-01297R1The effect of musical training on the processing of audiovisual correspondences: Evidence from a reaction time taskPLOS ONE

Dear Dr. Ihalainen,

Thank you for submitting your manuscript to PLOS ONE. After careful consideration, we feel that it has merit but does not fully meet PLOS ONE’s publication criteria as it currently stands. Therefore, we invite you to submit a revised version of the manuscript that addresses the points raised during the review process.

We look forward to receiving your revised manuscript.

Kind regards,

Deborah Apthorp, Ph.D

Academic Editor

PLOS ONE

Journal Requirements:

Additional Editor Comments (if provided):

I apologise for the delay - Reviewer 2 declined to review again, and we felt it was important to source another opinion. Reviewer 3 was aware that the paper had been previously reviewed, and suggests minor revisions, in harmony with Reviewer 1.

In particular, please ensure that all data and code for this study is publicly available, in line with the policies of the journal.

Reviewers' comments:

Reviewer's Responses to Questions

**Comments to the Author**

1. If the authors have adequately addressed your comments raised in a previous round of review and you feel that this manuscript is now acceptable for publication, you may indicate that here to bypass the “Comments to the Author” section, enter your conflict of interest statement in the “Confidential to Editor” section, and submit your "Accept" recommendation.

Reviewer #1: (No Response)

Reviewer #3: (No Response)

2. Is the manuscript technically sound, and do the data support the conclusions?

Reviewer #1: Yes

Reviewer #3: Yes

3. Has the statistical analysis been performed appropriately and rigorously? 

Reviewer #1: Yes

Reviewer #3: Yes

4. Have the authors made all data underlying the findings in their manuscript fully available?

Reviewer #1: Yes

Reviewer #3: No

5. Is the manuscript presented in an intelligible fashion and written in standard English?

Reviewer #1: Yes

Reviewer #3: Yes

6. Review Comments to the Author

Reviewer #1: I thank the authors for their very detailed responses and revision. In particular, the van Bastian & Druey paper was interesting, thanks for bringing that to my attention. Please note, though, that it does not appear in the reference list…

All the minor points have been addressed except one (although it was marked as corrected). In the Participants section, the second paragraph, beginning “We applied a standard…”, should be moved to be either the first or second paragraph of the Analysis section, depending on which the authors feel flows better. This is because it is more about prepping the data for analysis than it is about who participated.

I noticed a few errors of English expression in the revision: for example, “our main interest lied” rather than “lay”; and also some typos, for example, “thank” instead of “than”. Please make sure these, and any others, are caught at the proof stage.

Apart from these small items, it looks great!

Reviewer #3: I had the pleasure of reading this interesting manuscript about crossmodal integration in musicians and nonmusicians. As I read this for the first time, I will give a general overview of the manuscript, and then add specific comments. Note that I also read the previous reviews and it seems to me that the authors addressed well all the points raised. My comments will be, however, slighlty different from those already mentioned, but will not require any substantial modification of the present manuscript.

First of all I believe that the study itself is well designed, the manuscript reads well and the theoretical background and discussion are sufficiently rich. I have a general advice though: throughout the manuscript, we can often read "the effects of music training": note that very few studies could really prove that the music training causes some improvements in various perceptual/cognitive skills. Most of studies are based on the comparison of adult musicians and nonmusicians. This is not sufficient to talk about cause/effect relationships. I encourage the authors to talk about "association with the music training" instead of "effects". Then, in the discussion, clearly there is space of mentioning why the authors believe it is reasonable to consider the music training as the cause. But it is still an intepretation, and the authors should acknowledge, perhaps in the limitation section, that the present study cannot infere any cause-effect relationship.

This is particularly true (and here I suggest to add another limitation) as there was no general control of cognitive abilities of the two groups. For example, the speed of processing subtests of the WAIS-IV could have been informative in explaining why there were no differences between groups in the RTs. Having no control tasks, cannot exclude that the two groups were different in terms of general cognitive abilities. Also, I did not read any information about years of education. Was this variable collected? Are the group different in years of education?

Again on the groups differences. I see that the inclusion criteria for nonmusicians was not having formal training outside from the mandatory classes at school. Did the authors checked whether the nonmusicians could have, anyway, learnt to play an instrument as self-taught? For instance, as they "self-identified as having no musical expertise", was musical expertise defined as having received formal lessons? This is important, because one could still play a bit as an amateur but consider him/herself not an expert. What were the exact questions asked to gather information about musical expertise? Similarly, I find the range of musical expertise in the musician group quite large. Studies including musicians usually have stricter criteria (e.g., 7-8 years of training at least). 3 years seems not a enough. Did the authors check at least whether the musicians were active at the moment of the testing? Because if an individual had 3 years of experience but stopped to play 10 years before, well, this wouldn't really qualify as being a musician in my opinion.

I encourage the authors to add more details about the two groups, and if these details are not available, to include the criteria used to create the two groups as a possible limitation (that might also explain different results from previous studies, perhaps). I know that the athors provide already many analyses, but perhpas it could be interesting to look at the correlations between years of musical expertise and RTs/accuracy, as the range is very wide?

Finally, I do not see in the manuscript any statement or link for data availability. My apologies if this will appear later on, in any case, I think it would be great to provide a link where readers can access the dataset.

Some specific details I noticed:

-Figures: Figure 1 and 2 seem very low resolution.

-page 4, line 85: "audiovisual congruency effects" I suggest adding a definition of what these effects are in practice (e.g.,higher accuracy in identifying congruent pairs, shorter RTs, etc.?), as there might be different things to which the authors are referring.

-page 4, line 94: "at discriminating the stimuli" This reads a bit vague, I suggest clarifying what the task required.

-page 4, line 97,98: " pitch-elevation stimuli" and "non-symbolic magnitude" are not clear yet in the introduction, I suggest to define what are these types of stimuli, otherwise the reader will understand it only in the method section

-page 5, line 114: here the "detection reaction time task" is also a bit vague, had the participants to respond to congruency again?

-page 5, line 117: Apologise if it is my mistake, but by reading the description of the study by Bidelman et al., it seems that the musicians had less frequently the audiovisual illusion. Does this mean that they integrated better the stimuli? Because intuitively, I would say that if they integrated them better, they would suffer the illusion more, not less. Having less frequently the illusion (or with shorter - less detectable - durations) might indicate, to me, that they could segregate better the two types of stimuli, not integrate them. But I might misunderstand what it's written.

-Page 7, line 150: I'm not a native speaker but starting a paragraph with "therefore" reads strange.

- Page 7, line 162: I suggest writing that the power analysis is explained later on, otherwise at a first glance one could wonder why no details about it are reported.

-Page 8, line 170: "made a mistake". How is the mistake defined here? Because later on, mistakes (wrong answers) are taken into account in the analyses, so I believe that this is a different type of mistake.

-Page 17. I find it a bit strange to read that there is a difference between musicians and nonmusicians in numbers of mistakes in the incongruent trials, but then, in the last paragraph with this analysis there is no difference between groups in mistakes in congruent and incogruent conditions. (see line 382, "irrespective of musicianship"). Are these two results a bit in contraddiction? If the musicians have less mistakes in the incogruent conditions, I would expect an interaction, not an overall effect of congruency.

-page 19, line 410, 411: Can the authors report the statistics (at least the p-values, as before) for the post-hoc significant comparisons?

-page 20, line 416: I think that here mentioning "effects of long-term training" is quite tricky for the reasons mentioned before: (1) there is no way to understand any effect of musical training with the present study, (2) speaking about long training with a inclusion criteria of >3 years seems a bit optimistic).

I think that if these minor details are clarified, the manuscript will be then ready to be publicated.

7. PLOS authors have the option to publish the peer review history of their article (what does this mean?). If published, this will include your full peer review and any attached files.

Reviewer #1: No

Reviewer #3: No

---

## [Author Response · Author response to Decision Letter 1]

12 Feb 2023

Response to Reviewers

Reviewer #1 comments to the authors

I thank the authors for their very detailed responses and revision. In particular, the van Bastian & Druey paper was interesting, thanks for bringing that to my attention. Please note, though, that it does not appear in the reference list…

All the minor points have been addressed except one (although it was marked as corrected). In the Participants section, the second paragraph, beginning “We applied a standard…”, should be moved to be either the first or second paragraph of the Analysis section, depending on which the authors feel flows better. This is because it is more about prepping the data for analysis than it is about who participated.

I noticed a few errors of English expression in the revision: for example, “our main interest lied” rather than “lay”; and also some typos, for example, “thank” instead of “than”. Please make sure these, and any others, are caught at the proof stage.

Apart from these small items, it looks great!

Response to reviewer: We wish to thank the reviewer for the overall encouraging feedback on both revision rounds, and we do want to apologize for these typos and mishaps that had slipped into the manuscript. We have proof read the manuscript again and corrected the mistakes accordingly. Due to issues in switching to a different reference manager, we had failed to include one of the in-text references in the bibliography. This has now been corrected (added under ‘B’). We have also followed the suggestion of moving the paragraph from the Participants section to a more proper location (under Analysis.

Reviewer #3 comments to the authors 

I had the pleasure of reading this interesting manuscript about crossmodal integration in musicians and nonmusicians. As I read this for the first time, I will give a general overview of the manuscript, and then add specific comments. Note that I also read the previous reviews and it seems to me that the authors addressed well all the points raised. My comments will be, however, slightly different from those already mentioned, but will not require any substantial modification of the present manuscript.

First of all I believe that the study itself is well designed, the manuscript reads well and the theoretical background and discussion are sufficiently rich. I have a general advice though: throughout the manuscript, we can often read "the effects of music training": note that very few studies could really prove that the music training causes some improvements in various perceptual/cognitive skills. Most of studies are based on the comparison of adult musicians and nonmusicians. This is not sufficient to talk about cause/effect relationships. I encourage the authors to talk about "association with the music training" instead of "effects".

Then, in the discussion, clearly there is space of mentioning why the authors believe it is reasonable to consider the music training as the cause. But it is still an intepretation, and the authors should acknowledge, perhaps in the limitation section, that the present study cannot infere any cause-effect relationship.

This is particularly true (and here I suggest to add another limitation) as there was no general control of cognitive abilities of the two groups. For example, the speed of processing subtests of the WAIS-IV could have been informative in explaining why there were no differences between groups in the RTs. Having no control tasks, cannot exclude that the two groups were different in terms of general cognitive abilities. Also, I did not read any information about years of education. Was this variable collected? Are the group different in years of education?

Response to reviewer: The authors would like to sincerely thank the reviewer for the positive and very constructive and insightful feedback. We absolutely agree with the point regarding causality: it is indeed the case that we cannot demonstrate causality in a decision reaction-time task such as in the present manuscript. We do not wish to claim that such causality is demonstrated, although we do recognise that we have used unclear language and semantics that easily reads as we indeed wish to suggest such causal relationship. We have changed the language throughout the manuscript to refer to relationship/association/link between musical training and advantages in multisensory integration, as suggested. We also added a paragraph in the discussion section stating this explicitly. In that paragraph we also suggest that future studies should use a control task/tasks and collect more extensive background knowledge from the participants to have stronger grounding for suggesting causality between musicianship and observed advantages (lines 568-576).

Again on the groups differences. I see that the inclusion criteria for nonmusicians was not having formal training outside from the mandatory classes at school. Did the authors checked whether the nonmusicians could have, anyway, learnt to play an instrument as self-taught? For instance, as they "self-identified as having no musical expertise", was musical expertise defined as having received formal lessons? This is important, because one could still play a bit as an amateur but consider him/herself not an expert. What were the exact questions asked to gather information about musical expertise? 

Response to reviewer: This is again an important point. We did ask for any musical experience from the participants, and as we mention in Participants section under Materials and Methods, we disregarded one participant due to having been playing the piano between the ages of 7 to 12 years old. Self-identified musical experience here was based on asking if the participants played any instruments, composed music on a computer, or had taken any lessons for any instruments at any point in their life. Any continuous lessons or practicing excluding the compulsory music lessons in elementary/high-school resulted in exclusion from the experiment. To be more precise, we have added this last sentence into the manuscript (lines 181-182). 

Similarly, I find the range of musical expertise in the musician group quite large. Studies including musicians usually have stricter criteria (e.g., 7-8 years of training at least). 3 years seems not a enough. Did the authors check at least whether the musicians were active at the moment of the testing? Because if an individual had 3 years of experience but stopped to play 10 years before, well, this wouldn't really qualify as being a musician in my opinion. I encourage the authors to add more details about the two groups, and if these details are not available, to include the criteria used to create the two groups as a possible limitation (that might also explain different results from previous studies, perhaps).

Response to reviewer: We indeed did ask whether the musicians were currently active and that was one of the inclusion criteria and we have added this detail into the manuscript as well. We also asked the participants their mean training time per week, which varied between 1 to 6 hours, but as we didn’t have this information for all of the musicians, we did not include it in the manuscript. Moreover, only 2 musicians reported having less than 6 years of training, hence we do not believe the more flexible definition of musicianship played a significant role in the results (in comparison to other studies in the field). We have added more details to the manuscript regarding the definition of musicianship.

We also noticed a mistake in the mean number of musical education and updated the value to the correct mean after participant exclusion.

I know that the athors provide already many analyses, but perhpas it could be interesting to look at the correlations between years of musical expertise and RTs/accuracy, as the range is very wide?

Response to reviewer: As the manuscript already includes a number of analyses, we are hesitant to add anymore into the mix. We also feel that our analyses already cover these questions to an extent: it was precisely our hypothesis that reaction times and number of mistakes made would decrease as a function of musical expertise which could be captured with number of years of training. 

However, based on this feedback we did run the correlation analyses suggested. All correlations with reaction times were non-significant, while all correlations with the number of mistakes made in each category were statistically significant with all correlations being negative (i.e. less mistakes made when musical education increased). This is not surprising given our previous results indicating that musicians made fewer mistakes than non-musicians (with training of 0 years), and that no differences were found in reaction times.

These results remained the same regardless of whether we correlated only musicians or included non-musicians (with training years of zero) into the sample. Hence, respectfully, we do not think including them would add a lot of value into the manuscript.

Finally, I do not see in the manuscript any statement or link for data availability. My apologies if this will appear later on, in any case, I think it would be great to provide a link where readers can access the dataset.

Response to reviewer: During the submission process we mentioned that the data will be made available upon publication. It is now online and can be found at https://gin.g-node.org/rihalai/Crossmodal_Correspondences

Some specific details I noticed:

-Figures: Figure 1 and 2 seem very low resolution.

Response to reviewer: We agree figure 1 had low resolution and we have re-made it with higher resolution. The resolution in figure 2 is better and is dictated by the resolution of the original stimuli images.

-page 4, line 85: "audiovisual congruency effects" I suggest adding a definition of what these effects are in practice (e.g.,higher accuracy in identifying congruent pairs, shorter RTs, etc.?), as there might be different things to which the authors are referring.

Response to reviewer: These studies are discussed in detail in the very next paragraph (starting from line 87). This was not very clear, though, and we have now edited the manuscript such that the new paragraph starts from the line first mentioning the “audiovisual congruency effects” and is followed by the more detailed discussion of those experiments.

-page 4, line 94: "at discriminating the stimuli" This reads a bit vague, I suggest clarifying what the task required.

Response to reviewer: Agreed. We changed this to refer to the task of judging whether the stimuli were congruent with the rule.

-page 4, line 97,98: " pitch-elevation stimuli" and "non-symbolic magnitude" are not clear yet in the introduction, I suggest to define what are these types of stimuli, otherwise the reader will understand it only in the method section

Response to reviewer: We have edited the text to refer to spatial elevation, which is mentioned before, and generally to visual representations of magnitudes (in conjunction with pitch).

-page 5, line 114: here the "detection reaction time task" is also a bit vague, had the participants to respond to congruency again?

Response to reviewer: We have added more details of the task into the description. It now says “…with a detection reaction time task, in which the participants were instructed to click a mouse button immediately upon perception of auditory, tactile, or simultaneous audio-tactile stimuli, and reported that…” (line 114-118).

-page 5, line 117: Apologise if it is my mistake, but by reading the description of the study by Bidelman et al., it seems that the musicians had less frequently the audiovisual illusion. Does this mean that they integrated better the stimuli? Because intuitively, I would say that if they integrated them better, they would suffer the illusion more, not less. Having less frequently the illusion (or with shorter - less detectable - durations) might indicate, to me, that they could segregate better the two types of stimuli, not integrate them. But I might misunderstand what it's written.

Response to reviewer: This is a fair question. We discuss Bidelman et al. (2016) paper as an example of observed processing speed advantage in musicians over non-musicians, and this was indeed one of the results of Bidelman et al. (i.e., musicians were faster at making their response than nonmusicians and that musicians were not only more accurate at processing concurrent audiovisual cues but considerably faster at judging the composition of audiovisual stimuli). 

In addition, they did observe musicians to have lower susceptibility for perceiving the illusory effect. In the paper, Bidelman et al. talk about “considerably more refined binding of auditory and visual cues” and “..musicians have enhanced multisensory integration and are better able to accurately parse audiovisual cues”. They conclude that musical experience “improves multimodal processing and integration of multiple sensory systems”. On the other hand, in their paper, they cite other results showing age-related increased multisensory integration “.. as evidenced by broader temporal binding window”. We take it that it is their position, that taken together, these results indicate a more refined, improved multisensory integration/processing (whereas looking at the narrower temporal binding window in musicians alone would probably be characterized differently if discussed in isolation).

In the manuscript, we were careful not to talk about “increased multisensory integration” in relation to these results, but rather, “improved multisensory integration”. We have further edited the part to refer to a more refined, improved integration of multiple sensory systems in a domain-general manner.

-Page 7, line 150: I'm not a native speaker but starting a paragraph with "therefore" reads strange.

Response to reviewer: We have changed this to “Hence”.

- Page 7, line 162: I suggest writing that the power analysis is explained later on, otherwise at a first glance one could wonder why no details about it are reported.

Response to reviewer: Corrected.

-Page 8, line 170: "made a mistake". How is the mistake defined here? Because later on, mistakes (wrong answers) are taken into account in the analyses, so I believe that this is a different type of mistake.

Response to reviewer: These mistakes were indeed wrong answers (analysed later on). Here, we refer only to the reaction time measure: for the calculations of the reaction times, we only used the recorded data from the correct responses. Note that this paragraph is now moved under the Analysis-section, as per the suggestion of Reviewer #1.

-Page 17. I find it a bit strange to read that there is a difference between musicians and nonmusicians in numbers of mistakes in the incongruent trials, but then, in the last paragraph with this analysis there is no difference between groups in mistakes in congruent and incogruent conditions. (see line 382, "irrespective of musicianship"). Are these two results a bit in contraddiction? If the musicians have less mistakes in the incogruent conditions, I would expect an interaction, not an overall effect of congruency.

Response to reviewer: Here, the last result did not look at group-differences at all (between musicians and non-musicians). Rather, we analysed the overall number of mistakes made (musicians and non-musicians combined) in all congruent trials and in all incongruent trials, and found a large difference indicating more mistakes in incongruent trials (as expected). This difference was there irrespective of musicianship (i.e. for both groups combined). We can understand that by mentioning musicianship here may be confusing as we are looking at all participants together, and hence, we have removed that part from the paragraph (“irrespective of musicianship”).

-page 19, line 410, 411: Can the authors report the statistics (at least the p-values, as before) for the post-hoc significant comparisons?

Response to reviewer: We have added the p-values for this estimated marginal means interaction. We have also corrected the language in this paragraph, and related the interaction more directly to elevation category.

-page 20, line 416: I think that here mentioning "effects of long-term training" is quite tricky for the reasons mentioned before: (1) there is no way to understand any effect of musical training with the present study, (2) speaking about long training with a inclusion criteria of >3 years seems a bit optimistic).

Response to reviewer: This is now corrected to “.. explore the association between long-term musical training and bimodal sensory integration”.

I think that if these minor details are clarified, the manuscript will be then ready to be publicated.

Response to reviewer: We appreciate all the feedback and feel that the manuscript is much stronger after editing it accordingly.

---

## [Editor Report · Decision Letter 2]

22 Feb 2023

The relationship between musical training and the processing of audiovisual correspondences: Evidence from a reaction time task

PONE-D-22-01297R2

Dear Dr. Ihalainen,

We’re pleased to inform you that your manuscript has been judged scientifically suitable for publication and will be formally accepted for publication once it meets all outstanding technical requirements.

Kind regards,

Deborah Apthorp, Ph.D

Academic Editor

PLOS ONE
---

## [Editor Report · Acceptance letter]

27 Feb 2023

PONE-D-22-01297R2 

The relationship between musical training and the processing of audiovisual correspondences: Evidence from a reaction time task 

Dear Dr. Ihalainen:

I'm pleased to inform you that your manuscript has been deemed suitable for publication in PLOS ONE. Congratulations! Your manuscript is now with our production department. 

Kind regards, 

on behalf of

Dr. Deborah Apthorp 

Academic Editor

PLOS ONE